



# Thermal structure of the Amery Ice Shelf from borehole observations and simulations

Yu Wang[1,2], Chen Zhao[1], Rupert Gladstone[3], Ben Galton-Fenzi[1,4], Roland Warner[1]

[1]Australian Antarctic Program Partnership, Institute for Marine and Antarctic Studies, University of Tasmania, Hobart, Australia
[2]Ocean University of China, Qingdao, China
[3]Arctic Centre, University of Lapland, Rovaniemi, Finland
[4]Australian Antarctic Division, Kingston, Australia

*Correspondence to*: Yu Wang (yu.wang0@utas.edu.au)

**Abstract.** The Amery Ice Shelf (AIS), East Antarctica, has a layered structure, due to the presence of both meteoric and marine ice. In this study, the thermal structure of the AIS and its spatial pattern are evaluated and analysed through borehole observations and numerical simulations with Elmer/Ice, a full-Stokes ice sheet model. In the area with marine ice, a near-isothermal basal layer up to 120 m thick is observed, which closely conforms to the pressure-dependent freezing temperature of seawater. In the area experiencing basal melting, large temperature gradients, up to -0.36 ℃ m⁻¹, are observed at the base. Three-dimensional (3-D) steady-state temperature simulations with four different basal mass balance datasets reveal a high sensitivity of ice-shelf thermal structure to the distribution of basal mass balance. We also construct a one-dimensional (1-D) temperature column model to simulate the process of ice columns moving along flowlines with time-evolving boundary conditions, which achieves slightly better agreement with borehole observations than the 3-D simulations. Our simulations reveal internal cold ice advected from higher elevations by the AIS's tributary glaciers, warming downstream along the ice flow, and we suggest the thermal structures dominated by the cold core ice may commonly exist among Antarctic ice shelves. For the marine ice, the porous structure of its lower layer and interactions with ocean below determine the local thermal regime and give rise to the near-isothermal phenomenon. The limitations in our simulations identify the need for ice shelf/ocean coupled models with improved thermodynamics and more comprehensive evaluation of boundary conditions. Given the temperature dependence of ice rheology, the depth-averaged ice stiffness factor $B(T_h)$ derived from the most realistic simulated temperature field is presented to quantify the influence of the temperature distribution on ice shelf dynamics. The full 3-D field of this factor will assist as an input to future modelling studies.

## 1 Introduction

The Amery Ice Shelf (AIS) (Fig. 1; ~70° S, 70° E) is the largest ice shelf in East Antarctica. It has an estimated floating ice area of 60,000 km² (Galton-Fenzi et al., 2008), extending more than 550 km from its southern grounding zone to the ice front in Prydz Bay. The thickest region of the ice shelf is at the southern grounding zone, with a thickness of ~2,500 m (Fricker,





2002). The AIS is fed primarily by the Lambert, Mellor and Fisher Glaciers, which account for 60.5% of the total ice mass flux (Yu et al., 2010). The remaining ice flux across the grounding line is contributed by other tributaries on the eastern and western sides of the AIS. The AIS together with its tributary glaciers and their catchments is referred to as the Lambert-Amery Glacial system (LAGs).

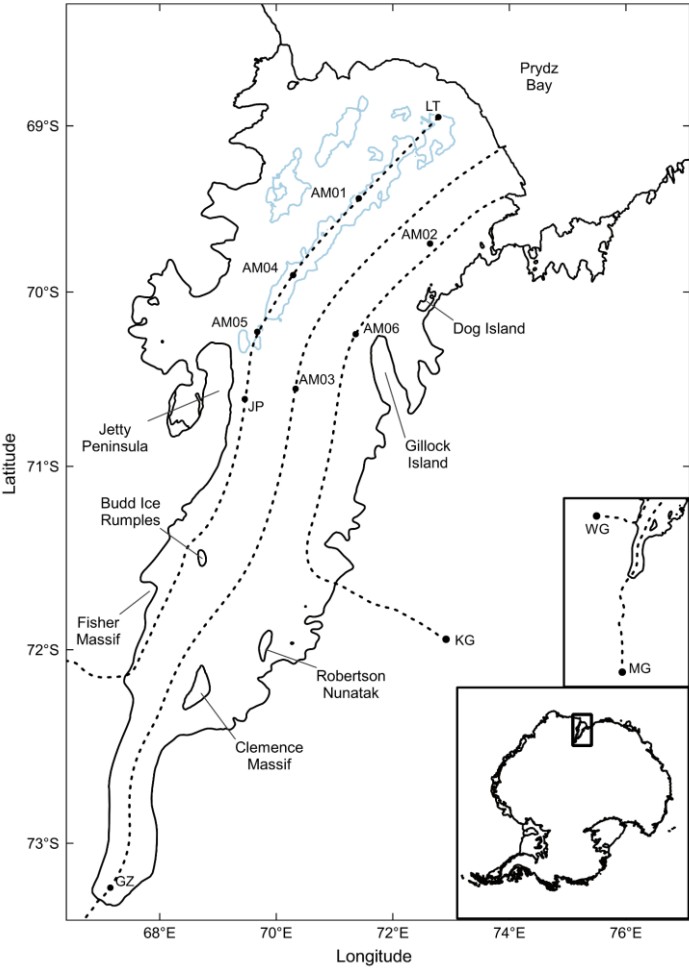

**Figure 1: The Amery Ice Shelf with significant features and AM01–AM06 borehole locations. Three dashed lines, derived from MEaSUREs InSAR-Based Antarctica Ice Velocity Map (Rignot et al., 2017), indicate the characteristic ice flowlines used in this study. The Jetty Peninsula flowline (henceforth JP flowline) starts from what we term the West Tributary Glacier (WG) and passes**

**through Jetty Peninsula point (JP), AM05, AM04 and AM01 boreholes, ending at the "Loose Tooth" point (LT). The terminology of the flowline points JP and LT follows Craven et al. (2009), but the specific locations are slightly different. The AM03 flowline originates from Mellor Glacier (MG), passes through Grounding Zone point (GZ) and AM03 borehole, ending at the ice front. The AM06 flowline originates from Kronshtadtskiy Glacier (KG), passes through AM06 borehole and passes close to AM02 borehole. Marine ice band with thickness greater than 100 m is shown with the light blue contours (Fricker et al., 2001). The locations of the**



**grounding mask and the ice front are from Depoorter et al. (2013). Insets show the origins of the JP and AM03 flowlines, and location of the Amery Ice Shelf in East Antarctica.**

The marine ice (i.e., basal ice formed from ocean water) layer under the AIS is an important feature of its overall structure, which could stabilize the ice shelf (Khazendar et al. 2009, Kulessa et al. 2014). Based on satellite radar altimeter and airborne
radio-echo sounding (RES) measurements, Fricker et al. (2001) derived the spatial distribution of marine ice layer under the AIS. The thickness of basal marine ice was estimated to be as great as 190 m (Fricker et al. 2001), while borehole measurements revealed that the thickness exceeds 200 m (Craven et al., 2004, 2005, 2009). Most of the marine ice is located in two longitudinal zones in the north-western AIS and extends along ice flowlines all the way to the ice front (light blue contours in Fig. 1; after Fricker et al., 2001).


The overturning ocean circulation under the ice shelf, together with changes in the in situ freezing point of seawater, contribute to the refreezing process and the formation of marine ice (Lewis & Perkin 1986). Through the observations obtained from borehole video cameras and instrument moorings (Craven et al. 2005, Craven et al. 2014), the formation processes of the marine ice layer under the AIS and its structure have been preliminarily revealed. Frazil ice accretes and platelets consolidate
at the ice–ocean interface, forming the original basal marine ice (Lambrecht et al. 2007, Craven et al. 2014, Herraiz-Borreguero et al. 2013, Galton-Fenzi et al. 2012). The newly formed marine ice, which is highly porous and hydrologically connected to the ocean below, slowly consolidates and undergoes a pore closure process (Craven et al. 2009). During the hot-water drilling at AM01 and AM04, a sudden change of the water level in the borehole indicated that drilling had established a hydraulic connection between the water filled borehole and the ocean beneath the ice shelf well above the actual ice shelf base (Craven
et al. 2004, Craven et al. 2009). The remaining porous ice was still mechanically strong and had to be removed by continued drilling. The hydraulic connection depths are regarded as the interface between upper impermeable and lower permeable marine ice layers (Craven et al. 2009). Craven et al. (2009) also indicated that the cavities between the platelets account for more than 50% of the total volume in the deepest marine ice, while the porosity of the impermeable marine ice is much lower. Due to the porous structure of the deeper marine ice layer, together with the presence of meteoric ice (i.e., ice formed from
compacting snow) flowing from the continent and also deposited on the ice shelf, and the surface firn layer (Treverrow et al. 2010), the AIS has a layered vertical structure, which will be explored in this study.

The knowledge of the thermal structure of ice shelves is of high practical interest, and the internal temperature regime records the past climate and thermal conditions upstream (Humbert, 2010). Many studies have been carried out on the thermal structure
of the ice sheets/glaciers (e.g., Jania et al., 1996; Ryser, 2014; Saito and Abe-Ouchi, 2004; Seroussi et al., 2013) and ice shelves (e.g., Budd et al., 1982; Craven et al., 2009; Humbert, 2010; Kobs et al., 2014). To explore the vertical temperature regime of ice shelves, hot-water drilling is commonly used to access the ice shelf interior (e.g., Craven et al., 2004; Makinson, 1994). The internal temperature of ice shelves can be obtained through long-term temperature measuring instruments deployed within





the boreholes. Thermistor strings with surface loggers can provide long-term point borehole temperature at different depths.

In a modelling study, Humbert (2010) evaluated the thermal regime of the Fimbulisen (Fimbul ice shelf) based on thermistor data from a single borehole, which showed a cold middle part inside the ice shelf. However, these point sensors are not able to provide spatially continuous temperature measurements and the vertical resolution is limited by the number of thermistors. The fibre-optical temperature sensing, also known as distributed temperature sensing (DTS), is a better approach to achieve continuous in situ temperature measurements inside an ice shelf (Tyler et al. 2013). Among Antarctic ice shelves, DTS

deployments were first made in the AM05 and AM06 boreholes of the AIS (Warner et al., 2012). Kobs et al. (2014) derived the temperature gradient at the ice–ocean interface of the McMurdo Ice Shelf from high-resolution DTS data, and also estimated seasonal basal melting using the evolution of the temperature gradient.

In the early stage of studies on the thermal regime of ice shelves, Wexler (1960) and Crary (1961) quantified the observed

temperature profiles at sites on the Ross Ice Shelf and derived steady-state solutions for the profiles, which is a function of basal melt rate. The earliest vertical temperature profile for the AIS was determined by measurements in the upper 320 m of the borehole G1 (69.44° S, 71.42° E; Budd et al., 1982), the same geographic location as the later AM01 site (Fig. 1). By fitting the measured temperature profile with the 1-D advection-diffusion equation, small temperature gradients were found within ~100 m of the upper and lower surfaces and the transition temperature gradient in between is stable and relatively larger (Budd

et al. 1982). The temperature profiles obtained within the marine ice band at AM01 and AM04 borehole sites also showed similar profile patterns (Craven et al. 2004, Craven et al. 2009). Craven et al. (2009) attributed the near-isothermal phenomenon of the bottom permeable layer to the accretion of marine ice as deposition of frazil ice platelets and suggested there is no conductive heat flux into the ice shelf from the ocean cavity.

However, the thermal structure of Antarctic ice shelves is not yet fully explored due to technical and logistical difficulties of internal ice shelf measurements. In this study, we focus on the thermal structure along three characteristic flowlines on the AIS in the areas with and without basal marine ice (Fig. 1) and compare the measured and simulated thermal structure of the AIS at six borehole sites (AM01–AM06). The temperature profiles are obtained from borehole thermistor strings at AM01–AM04 and DTS at AM05–AM06. A full-Stokes ice sheet model, Elmer/Ice (Gagliardini et al. 2013), is used to simulate the ice shelf

dynamics and the three-dimensional (3-D) steady-state temperature fields using four different basal mass balance datasets for the AIS. To further illustrate the formation and evolution of the vertical thermal regime, one-dimensional (1-D) temperature column simulations are designed to reconstruct the progress of ice columns moving along the flowlines with appropriately varying boundary conditions. We present the measured borehole temperatures in Sect. 2.1. The 3-D steady-state temperature simulations and 1-D temperature column simulations are introduced in Sect. 2.2 and Sect. 2.3, respectively. We present the

corresponding results in Sect. 3 and discuss them in Sect. 4 before giving the conclusions in Sect. 5.



## 2 Data and Methods

### 2.1 Borehole Temperature Measurements

The Amery Ice Shelf Ocean Research (AMISOR) project was launched to investigate ice–ocean interaction processes, the interaction with the interior grounded ice sheet and the properties of oceanic water masses beneath the ice shelf. (Allison,

2003). As a part of the AMISOR project, from 2001 onwards six boreholes, named AM01–AM06 (Fig. 1) were hot-water drilled on the AIS (Craven et al., 2014). Sites AM01, AM04 and AM05 are located on approximately the same ice flowline where basal marine ice is present, and we name it Jetty Peninsula flowline (hereafter JP flowline for simplicity) in this study. It originates from what we term the West Tributary Glacier (WG) of the AIS (Fig. 1). Sites AM02, AM03 and AM06 are in areas without basal marine ice where we determine another two characteristic flowlines (Fig. 1). The AM03 flowline originates

from Mellor Glacier (MG), passing through AM03 borehole. The AM06 flowline is from Kronshtadtskiy Glacier (KG), passes through AM06 borehole and passes close by AM02 borehole. After hot-water drilling processes, the boreholes were kept open for several days to make observations in the ocean cavity and deploy oceanographic mooring instruments (Craven et al., 2004), and thermistor strings or optical fibres, for long-term measurements. Two thermistor strings were deployed within and through each of the earlier boreholes, AM01–AM04. One was used to measure the internal ice shelf temperature and the other was for

tracking the ice–ocean interface. All the internal thermistor data points are used in this study, while only a few characteristic thermistor data points at the ice–ocean interface are selected, since the interface thermistors are closely distributed along the cable and the numerical difference between those thermistors is insignificant. The Sensornet Oryx instruments (distributed temperature sensors) and the optical fibres were deployed at AM05 and AM06 sites, which provided continuous profiles of the temperature distribution along the fibre cable with a spatial sampling interval of 1.015 m. Temperatures within the top 10

m of the firn layer were recorded by Climatological Automatic Weather Stations (AWS) of the Australian Antarctic Programme at AM01, AM02 boreholes and at the Amery G3 site (70.891° S, 69.871° E), which is approximately 42 km south of AM03, and these showed clear seasonal signals. Similar signals were also observed within 10 m of the firn layer on the McMurdo Ice Shelf (Kobs 2014). To eliminate the seasonal signals and derive "steady-state" vertical temperature profiles at the borehole sites for comparison with the simulations, some temperature data near the top surface at AM01–AM04 have been carefully

selected and averaged. Temperatures at 10 m depth at AM01 and AM02 are averaged from the corresponding AWS records, while the near surface temperatures at AM03 and AM04 are estimated with reference to all the available AWS data and a multi-year averaged surface temperature field (Comiso, 2000). Similarly, at AM05 and AM06, the DTS data within 20 m of surface is not considered, due to strong seasonal signals. Details about the temperature data, including the depths of the measuring instruments, are presented in Table 1.




**Table 1: Details of the borehole thermistor data at sites AM01-AM04 and DTS data at sites AM05 and AM06 in the AIS. The DTS conducts continuous temperature measurement with a spatial sampling interval of 1.015 m. Temperatures at sampling depths marked in parentheses are estimated using the available AWS data, and surface temperature field (Comiso, 2000) or in situ pressure melting temperature as appropriate; underlined depths are within the marine ice layer.**

| Sites | Locations during initial drilling | Total ice thickness during initial drilling (m) | Temporal coverages | Sampling depths of temperature measurement (m) | Accuracy (°C) |
|---|---|---|---|---|---|
| AM01 | 69.442° S, 71.417 ° E (Jan–2002) | 479 | 14/12/2003– 13/06/2004 | 10, 95, 215, 265, 315, 345, 365, 405, 460, 476, 480 | 0.01 |
| AM02 | 69.713° S, 72.640° E (Jan–2001) | 373 | 01/02/2003– 24/12/2007 | 10, 80, 150, 357, 373 | 0.1 |
| AM03 | 70.561° S, 70.332° E (Dec–2005) | 722 | 03/12/2006– 09/04/2007 | (3), 152, 202, 252, 302, 352, 452, 542, 632, (722) | 0.01 |
| AM04 | 69.900° S, 70.290° E (Jan–2006) | 603 | 17/04/2006– 19/04/2006 | (10), 80, 160, 240, 320, 400, 480, 500, 520, 550, 560, (603) | 0.01 |
| AM05 | 70.233° S, 69.675° E (Dec–2009) | 624 | 19/01/2012– 15/04/2012 | 20–624 | 0.02–0.2* |
| AM06 | 70.246° S, 71.364° E (Dec–2009) | 607 | 3-30/06/2012 1-31/12/2012 | 20–607 | 0.02–0.2* |

* Attainable DTS accuracy for the internal temperatures varies from 0.2 °C around -20 °C to 0.02 °C around -2 °C. This variation is due to the availability of accurate in situ calibration data.

After the temperature measuring instruments are deployed, the water in the boreholes refreezes in a relatively short time, while the borehole temperatures take a much longer time to fall back to equilibrium. This cooling process can be detected within

each borehole, assuring thermal disturbance produced by the drilling has basically dissipated and the borehole thermal regime is in approximate equilibrium. The internal thermistors at AM01 recorded a rapid dropping of borehole temperatures during 15 days after instruments were deployed, and they were still slowly decreasing at a rate of ~-0.04 °C day$^{-1}$ at 40 days, illustrating the long-term adjustment to the original temperature regime before hot-water drilling. The temperature observations at AM01-AM04 sites lasted for several years until 2009. However, during this period, battery exhaustion and surface logger failures

resulted in multiple measurement interruptions, and individual thermistor failures also led to the loss of data in space over the long-term measurement. Since January 2010, the distributed temperature sensors recorded borehole temperature along the optical fibres at AM05 and AM06 until 2013, but data gaps in these time series also exist due to operational difficulties. The non-equilibrium data disturbed by drilling work and the data with errors due to equipment failures are eliminated in this study. The temporal coverages of near-equilibrium temperature data used are listed in Table 1. A frozen-in device, either thermistor

string or distributed temperature sensor, is a "Lagrangian" measuring instrument, advected horizontally and vertically by the





flow of the ice shelf. The locations of the six boreholes during the initial drilling process are used for the subsequent analysis in this study, presented in Table 1.

## 2.2 The 3-D steady-state temperature simulations

The Elmer/Ice model (Gagliardini et al. 2013), a finite-element, full-Stokes ice sheet and ice shelf model, is used to simulate the ice flow for the whole LAGs, and to derive the corresponding steady-state temperature distribution within the ice. In simulating the ice flow dynamics the current study uses the adjoint inverse method to optimise both basal resistance and ice viscosity. More specifically, the optimised dimensionless basal resistance parameter, $\beta$ (Fig. 2a), governs the basal sliding relation:

$$\tau_b = C_b 10^\beta u_b \tag{1}$$

where $\tau_b$ is basal resistance, $u_b$ is sliding speed and $C_b$ is a basal resistance coefficient of 1 MPa m$^{-1}$ a. The optimised viscosity enhancement factor, $E_\eta$ (Fig. 2b), varies the viscosity, $\eta$ of the deforming ice, from that derived from Glen's flow law (Glen, 1958; Paterson, 1994):

$$\eta = \frac{1}{2} E_\eta^2 \, A(T_h)^{-1/n} \dot{\varepsilon}_e^{\frac{(1-n)}{n}} \tag{2}$$

where $n$ is the exponent in Glen's flow law; $A(T_h)$ is the corresponding deformation rate factor, dependent on ice temperature

relative to the pressure melting point, $T_h$; $\dot{\varepsilon}_e$ is the effective strain rate. Values of $E_\eta$ greater than one indicate stiffer ice than predicted by Glen's law, while values between zero and one indicate softer ice.

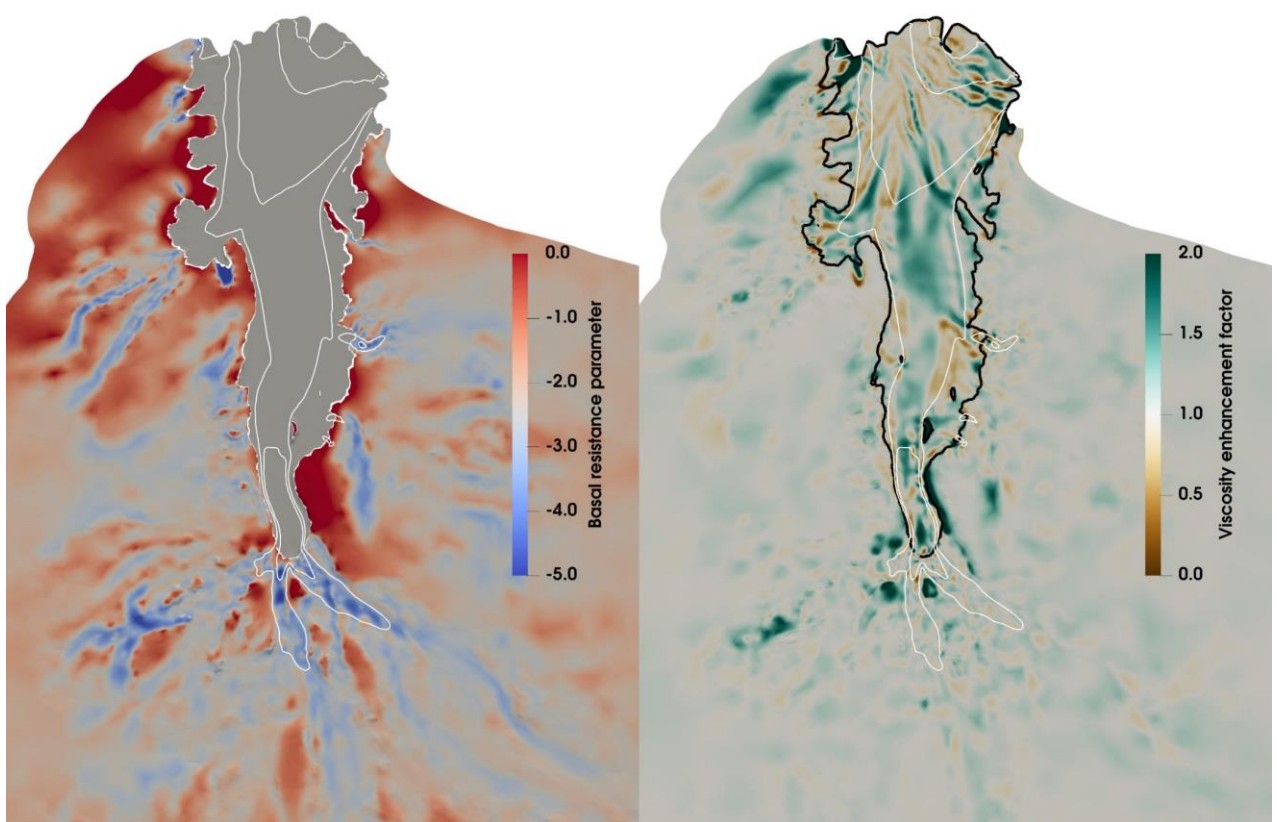

**Figure 2: The optimised basal resistance parameter $\beta$ (a) and viscosity enhancement factor $E_\eta$ (b) for the Lambert-Amery Glacial system in the 3-D model from Gladstone et al (2021, in preparation). The white lines represent the surface velocity contours of 200, 500, 1000, 1200 m a$^{-1}$, respectively, extracted from the inversion simulations. The black line in (b) represents the grounding line from BedMachine Antarctica (Morlighem et al., 2018).**

We introduce a temperature-dependent ice stiffness factor, $B(T_h)$, which will be discussed in the following sections, given as

$$B(T_h) = A(T_h)^{-1/n} \qquad (3)$$

Then the viscosity can be expressed as

$$\eta = \frac{1}{2} E_\eta^2 B(T_h) \, \dot{\varepsilon}_e^{\frac{(1-n)}{n}} \qquad (4)$$

In this study we utilize diagnostic simulations of LAGs dynamics with Elmer/Ice by Gladstone et al. (2021, in preparation) that use the bedrock and ice geometry from BedMachine (Morlighem et al. 2018), a 3-D internal ice temperature distribution (Seroussi et al., 2020) generated with the SICOPOLIS model (Greve et al., 2020) and take the observed horizontal surface velocities (Rignot et al. 2017) as the optimisation target. These simulations, their optimised inversions for $\beta$ and $E_\eta^2$, and their settings (including boundary conditions, other input data, and an L-curve analysis to assess regularisation) are described in full by Gladstone et al. (2021, in preparation). Here, we take as our starting point the optimised model state of Gladstone et al.





(2021, in preparation) corresponding to the end of their experiment E3 (summarised here in Fig. 2). This case corresponds to
the conventional dynamical boundary conditions: a stress-free upper surface; basal conditions of tangential frictional stress
and vanishing normal velocity for grounded ice, and vanishing tangential stresses and normal stress balancing ocean pressure
for the ice shelf. These experiments are essentially diagnostic solutions of the Stokes equations but include the usual short
prognostic phase to permit surface relaxation. The original ice temperature field from the SICOPOLIS modelling is retained
throughout.


We carry out further inversions for $\beta$ and $E_\eta^2$, with different upper and lower surface boundary conditions for the LAGs domain
(see Sect. 2.2.1 and 2.2.2). We present four experiments using different basal boundary conditions, corresponding to four
different choices about sub-ice shelf melting/refreezing. Each experiment comprises a sequence of three simulations
summarised in Table 2. As discussed below, our formulation of the basal boundary conditions for the momentum balance
equations for the ice shelf directly incorporates the influence of basal melting or freezing. The steady-state temperature field
simulation is essentially to solve the advection-diffusion equation using the ice velocities from the new optimizations, and
includes the corresponding internal strain heating and frictional heating at the grounded bed. Thermal boundary conditions are
discussed below. Aside from the upper and lower surface boundary conditions, the inversion simulations in the current study
are identical to the drag and viscosity inversions of Gladstone et al. (2021, in preparation). Each simulation uses the optimised
parameters from the previous stage. Tikhonov regularisation parameters of $10^3$ and $10^4$ are used for drag and viscosity
inversions respectively in the current study.

**Table 2: Experimental design of the 3-D simulation**

| Experiment | Basal mass balance source (References) | Simulation | Simulation type |
|---|---|---|---|
| BMB_ISMIP6 | ISMIP6 parameterisation (Seroussi et al. 2020) | DI_ISMIP6 | Inversion for basal drag coefficient |
| | | VI_ISMIP6 | Inversion for enhancement factor |
| | | TM_ISMIP6 | Steady-state temperature field simulation |
| BMB_ROMS | ROMS ocean model (Galton-Fenzi et al. 2012) | DI_ROMS | Inversion for basal drag coefficient |
| | | VI_ROMS | Inversion for enhancement factor |
| | | TM_ROMS | Steady-state temperature field simulation |
| BMB_CAL | Flux divergence calculation (Adusumilli et al. 2020) | DI_CAL | Inversion for basal drag coefficient |
| | | VI_CAL | Inversion for enhancement factor |
| | | TM_CAL | Steady-state temperature field simulation |
| BMB_CAL2 | Modified CAL (This paper) | DI_CAL2 | Inversion for basal drag coefficient |
| | | VI_CAL2 | Inversion for enhancement factor |
| | | TM_CAL2 | Steady-state temperature field simulation |





### 2.2.1 Lower surface boundary conditions

For the steady-state temperature simulations, the spatial distribution of geothermal heat flux under grounded ice of the LAGs as estimated from airborne magnetic data (Martos et al., 2017), and frictional heating for sliding of grounded ice are also incorporated. The lower surface temperature of the ice shelf is specified (a Dirichlet condition) as the in situ pressure-dependent freezing temperature of seawater (using the ice shelf draft) with a salinity of 35 psu.

The basal mass balance of the AIS is the flux of basal melting or refreezing (marine ice accretion) (Galton-Fenzi et al. 2012). The basal mass balance is expected to have great influence on the vertical thermal regime of the ice shelf (Kobs et al. 2014, Craven et al. 2009, Humbert 2010), and one of the aims of our study is to explore the sensitivity of the ice shelf thermal structure to the basal mass balance. However, the larger-scale sub-ice-shelf environment is difficult to access, and it remains unfeasible to directly detect the basal mass balance of the AIS. Ice mass flux divergence calculations and numerical ocean
models have been used to indirectly evaluate the basal mass balance, and its distribution pattern has been inferred (e.g., Adusumilli et al., 2020; Galton-Fenzi et al., 2012; Wen et al., 2010).

To evaluate the sensitivity of the simulated 3-D temperature distribution to basal mass balance, we adopt four basal mass balance datasets (Table 2; Fig. 3) in the 3-D simulations. The first basal mass balance dataset, BMB_ISMIP6, is from ISMIP6
(Seroussi et al. 2020), the Ice Sheet Model Intercomparison Project for CMIP6 (6th Coupled Model Intercomparison Project). The "local quadratic melting parameterisation" (Jourdain et al., 2020) is used in the current study. This is the only basal mass balance dataset we use that does not feature refreezing. The second basal mass balance dataset, BMB_ROMS, is derived from the basal ice–ocean thermodynamics and frazil dynamics by Galton-Fenzi et al. (2012), using the Regional Ocean Modelling System (ROMS). The third, BMB_CAL, is from Adusumilli et al. (2020), derived using an ice flux divergence calculation
(assuming ice shelves are in steady state). However, the porous structure of marine ice and the deposition of frazil ice increase uncertainties in estimating the thickness of an ice shelf with basal marine ice by buoyancy using surface elevations, mainly due to the uncertainties of marine ice density. To explore the response of the simulated temperature field to a higher basal accretion rate in our fourth mass balance dataset, BMB_CAL2, we double the basal freezing rate of BMB_CAL, while keeping the melt rate the same (i.e., positive mass balance values are doubled while negative mass balance values are left unchanged).

For the ice shelf, solving the Stokes equations using the standard basal boundary condition of matching the normal stress to seawater pressure, and adjusting the viscosity enhancement factor to optimise the surface velocities to observations, would produce ice shelf basal velocity fields corresponding to a basal mass balance consistent with the observed flow and the ice shelf geometry. However, our aim here is explore the implications for the ice shelf temperature of prescribed basal mass
budgets. Accordingly, the basal mass balance is imposed in all simulations in the current study as a Dirichlet condition on the ice velocity, in the direction normal to the lower surface. Specifically, basal melting and freezing correspond to an outward

(approximately downward) and inward (approximately upward) velocity respectively at the lower boundary. This is equivalent to the assumption that the ice shelf is in steady state.

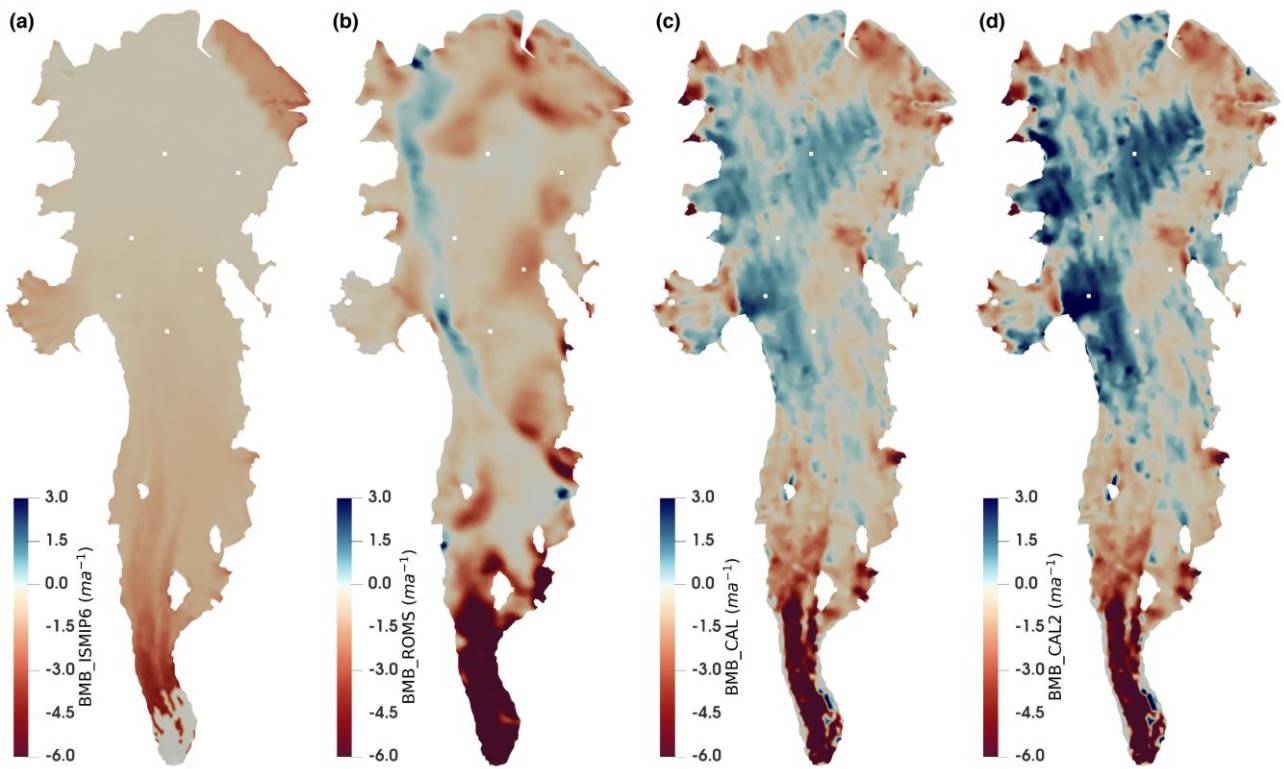

**Figure 3: The basal mass balance from (a) BMB_ISMIP6, (b) BMB_ROMS, (c) BMB_CAL and (d) BMB_CAL2. Negative implies basal melting, and positive implies freezing. White dots are the locations of AM01–AM06 boreholes, as shown in Figure 1.**

### 2.2.2 Upper surface boundary conditions

For the 3-D temperature simulations, the upper surface temperature is fixed by the mean surface air temperature field over 1979–1998 described in Comiso (2000). For all simulations, we also constrain the emergence velocity (the component of the velocity in the outward normal direction) at the upper surface using a non-zero resistance. A resistive normal stress, $\tau_r$, is applied in the direction normal to the upper surface:

$$\tau_r = -\boldsymbol{u} \cdot \boldsymbol{n}_s C_s \left[ 1 - \tanh \left( \frac{\|\boldsymbol{u}_{obs}\|}{u^*} \right) \right] \qquad (5)$$

where $\boldsymbol{u}$ is the modelled velocity, $\boldsymbol{n}_s$ is the outward unit normal vector at the ice surface, $C_s$ is a surface resistance coefficient, $u^*$ is a reference speed, and $\|\boldsymbol{u}_{obs}\|$ is the magnitude of the horizontal upper surface velocity from observations. We use surface resistance coefficient $C_s = 50$ MPa m$^{-1}$ a, and reference speed $u^* = 50$ m a$^{-1}$. This formulation imposes a stronger constraint on emergence velocity in regions where horizontal observed flow speeds are lower, mainly in the interior of the grounded ice sheet. In practice, the imposed normal stress is close to zero over the ice shelf and has very little impact on the dynamical



regime within the iceshelf. Further inland, it reduces a non-physical net downward advection of cold ice that occurs with the
more natural zero-stress upper surface boundary condition, thus giving a more plausible temperature regime over grounded

ice. The implications of non-zero emergence velocity, and the use of this boundary condition, are discussed in more detail by
Gladstone et al. (2021, in preparation).

## 2.3 The 1-D temperature column simulations

To explore the formation and evolution of the vertical thermal structure in the areas with and without basal marine ice, we also

conduct 1-D column simulations based on time-stepping to follow columns of ice along the JP flowline and AM03 flowline
(Fig. 1), respectively. A pioneering application of this approach was made by Macayeal and Thomas (1979) to interpret the
formation of the measured temperature profile at J9 borehole on the Ross Ice Shelf. In the current study, a series of key sites
along the two flowlines are determined as shown in Fig. 1 and Fig. 4. According to the spatial locations of key sites and the
surface velocity field of the AIS (Rignot et al. 2017), the time intervals between each site are derived as the key time stamps

in the simulations (Table 3).

**Table 3: Measured and estimated glaciological parameters and boundary conditions on the JP flowline and AM03 flowline for the 1-D temperature column simulations. Ice layer thicknesses at JP, AM04, AM01 and LT are from Craven et.al (2009). AM03 thickness comes directly from measurements during drilling; AM05 layers are estimated based on DTS data in this study (see Sect. 3.1). At**

**WG, MG and GZ thicknesses are derived from BedMachine Antarctica (Morlighem et al. 2018).**

| JP flowline | Ice thickness: meteoric, impermeable marine, permeable marine (m) | Length along flowline (km) | Time (years) | Surface temperature (°C) | Basal temperature (°C) | Vertical Strain rate ($a^{-1}$) |
|---|---|---|---|---|---|---|
| WG | 1500, (-), (-) | | | -25 | -1.3 | |
| to | | 286.4±10.7 | 2428.3±94.3 | | | -0.00019 |
| JP | 690, (-), (-) | | | -21.5 | -2.4 | |
| to | | 44.3±0.6 | 129.4±1.7 | | | -0.0029 |
| AM05 | 484, 83, 94 | | | -21.2 | -2.3 | |
| to | | 43.8±0.6 | 91.9±1.2 | | | -0.0028 |
| AM04 | 396, 137, 70 | | | -19.4 | -2.2 | |
| to | | 67.9±0.7 | 102.6±1.0 | | | -0.0048 |
| AM01 | 276, 100, 103 | | | -19.8 | -2.2 | |
| to | | 77.7±0.6 | 76.6±0.6 | | | -0.0084 |
| LT | 170, 66, 37 | | | -14.2 | -2.1 | |

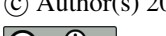



| AM03 flowline | Ice thickness (m) | Length along flowline (km) | Time (years) | Surface temperature (°C) | Basal temperature (°C) | Vertical Strain rate (a⁻¹) |
|---|---|---|---|---|---|---|
| MG | 3000 | | | -29 | -2.7 | |
| to | | 224.7±3.2 | 1338.8±17.1 | | | -0.00022 |
| GZ | 2500 | | | -21 | -3.8 | |
| to | | 323.6±3.4 | 758.7±7.8 | | | -0.00036 |
| AM03 | 722 | | | -19.5 | -2.4 | |

A vertical ice column with 100 equally spaced layers is constructed using Elmer/Ice. Each 1-D experiment consists of a series of simulations using temperature solvers in Elmer/Ice and including two stages: initial spin-up and forward transient simulations. Temperatures on the upper and lower surface are prescribed as boundary conditions, and come from a dataset of

Antarctic mean surface temperature over 1979-1998 (Comiso 2000) and in situ pressure melting point (or seawater freezing point for the ice shelf), respectively. The initial spin-up is a steady-state temperature simulation used to approximate the vertical thermal regime of grounded ice at the start of the flowline. We assume that the grounded ice has zero basal mass balance, while the upper surface is in positive mass balance and the value is extracted from RACMO2.3p2 (a regional atmospheric model) data (Melchior Van Wessem et al. 2018). Then the velocity of vertical advection $V$ at any depth $D$ due to surface

accumulation is assumed given by:

$$V(D)=(1-\frac{D}{H})A \tag{6}$$

where $H$ is the ice thickness and $A$ is the surface accumulation rate, as originally proposed by Robin (1955). We thus impose a vertical downward velocity on the upper surface and scale it linearly with depth to zero at the lower surface to approximate the vertical advection attributed to long-term surface accumulation.


Starting from the initial steady state, we perform a series of transient simulations representing evolution of the ice column as it is advected through the ice sheet and ice shelf as shown in Fig. 4 and Table 3. We impose boundary conditions, including the mass balances and temperatures on the upper and lower surfaces, taking 1-year time steps between the key time stamps. According to the location of the ice column at each time step, the surface mass balance is extracted from RACMO2.3p2

(Melchior Van Wessem et al. 2018). Similarly, the basal mass balance within the floating sector is extracted from Adusumilli et al. (2020) (corresponding to BMB_CAL) while zero basal mass balance is imposed for the grounding region. Ice thus flows vertically across the upper and lower surfaces of the column according to imposed mass balance (melting/freezing rates) in the transient simulations. The remaining boundary conditions are either linearly interpolated between key time stamps or directly specified for the intervals between them. The surface temperature at each time stamp is determined based on AWS observations

and surface air temperature dataset (Comiso 2000). The basal temperature at each time stamp is taken from borehole observations (where available) or calculated in situ pressure melting temperature. The ice thickness of the column model at

 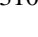



each key site is also fixed, based on borehole measurements and BedMachine Antarctica (Morlighem et al. 2018), while the vertical strain rates for each interval in Table 3 (always strain thinning) are selected to adjust the column thickness variations, in conjunction with imposed mass balance, to fit the prescribed ice thicknesses at the key sites.


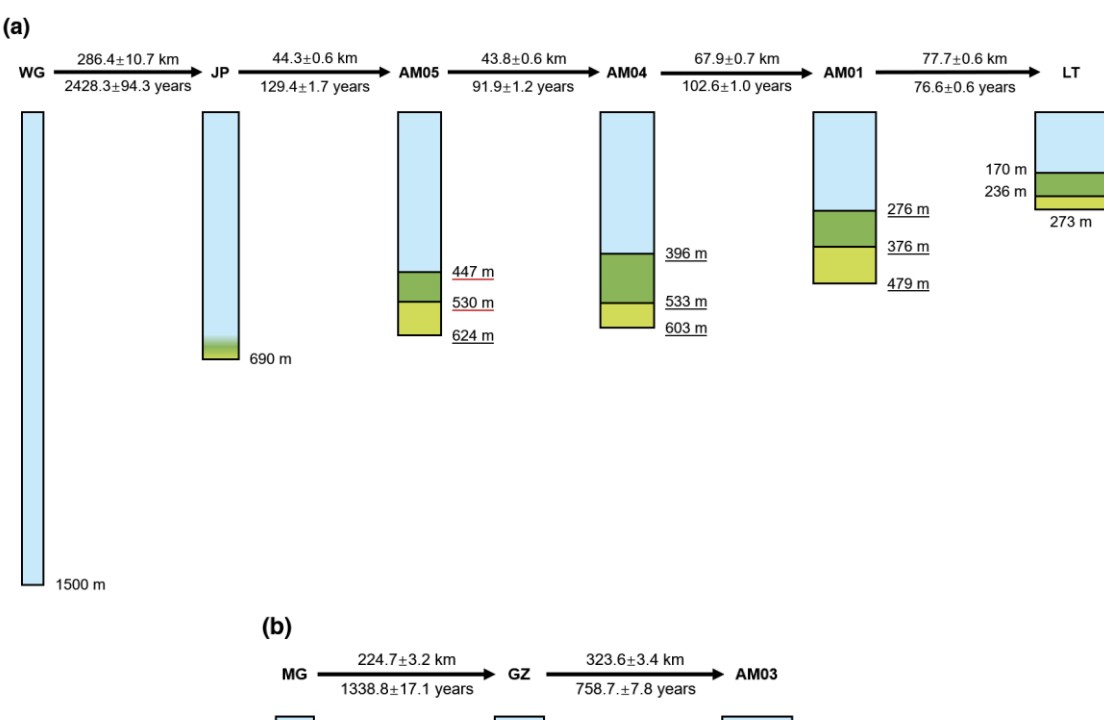

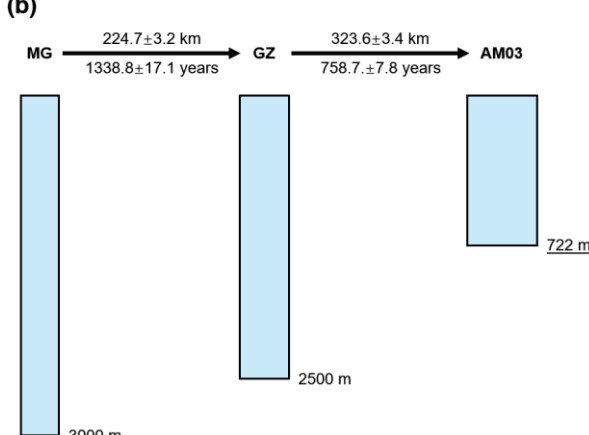

**Figure 4: Schematic diagrams of the 1-D temperature column simulation along (a) the JP flowline and (b) the AM03 flowline. From the top in (a): meteoric ice (blue); impermeable marine ice (dark green); and permeable marine ice (light green). At JP, there are**

**no observations indicating the condition of marine ice. The sources of ice layer thickness data are shown in Table 3. The increase in the width of the ice column qualitatively illustrates the strain thinning process along the flowlines.**





In general, the initial setup of the steady-state simulation and the boundary conditions in the transient simulation are inferred from a variety of available data, but there are still various uncertainties and assumptions. The 1-D simulations thus have great
flexibility in the choice of boundary conditions. However, they are also based on a series of assumptions:

1. The material properties of the ice column, including the density, heat capacity and conductivity, are constant everywhere and will not change with vertical strain process. The ice accreted on either surface in the transient simulation will also have the same material properties. Detailed physical parameters are shown in Table 4.

2. There is no horizontal shear in the ice column. The ice flow in our 1-D experiment is simply considered to be a plug flow. The ice column keeps vertical all the time in the simulations, and its height changes only by vertical strain and accretion/melting on both surfaces.

3. The ice flow of the AIS is assumed to be steady, such that ice streamlines of the horizontal velocity field coincide with
trajectories of columns of ice.

4. There is no thermal conduction in transverse direction, i.e., no heat flux through the lateral boundaries of the column, since the horizontal temperature gradient is considered to be much smaller than that in the vertical direction.

5. The basal temperature of grounded ice, as a Dirichlet condition, is always at pressure melting point of ice.

The column temperature regime can be extracted from the transient simulation at any temporal point, which corresponds to a certain spatial point on the flowline. Therefore, the simulations can be evaluated and optimized by comparing the simulated column temperature profile at the borehole sites with borehole measurements.


**Table 4: Standard physical parameters of the 1-D temperature column simulation. The heat capacity, $c_{ice}$, and thermal conductivity, $k_{ice}$, are functions of temperature, T, in K (Ritz, 1987).**

| Parameters | Symbol | Value | Units |
|---|---|---|---|
| Gravitational acceleration | $g$ | 9.81 | m s$^{-2}$ |
| Density of ice | $\rho_{ice}$ | 917 | kg m$^{-3}$ |
| Salinity of seawater | $S$ | 35 | psu |
| Heat capacity of ice | $c_{ice}$ | 146.3+7.253 $T$ | J kg$^{-1}$ K$^{-1}$ |
| Thermal conductivity of ice | $k_{ice}$ | 9.828 e$^{-5.7 \times 10^{-3} T}$ | W m$^{-1}$ K$^{-1}$ |

## 3 Results

### 3.1 Borehole thermal regime

The measured temperature profiles within the boreholes are relatively stable over the selected observation periods. The thermistors in AM01 and AM03 boreholes show a slight decrease in temperature within 0.05 °C in the temporal coverages (Table 1), which we attribute to continuing adjustment towards the original ice temperatures after hot-water drilling. The DTS time series in AM05 and AM06 suggest random fluctuations (noise) of ±0.05 °C in individual observations at any given depth,



which are reduced to ±0.01 °C by averaging consecutive measurements. This may be regarded as indicating the precision that
the DTS can provide. However, the averaging process also reveals small systematic differences between adjacent depths,
which points to limits on accuracy, although usually accuracy is limited by availability of precision calibration data. Borehole
thermal regimes are derived from averaging all the measurements (Table 1) selected to represent the thermal equilibrium. To
achieve continuous temperature profiles, spline interpolation is used to smooth discrete thermistor points to compare with DTS
profiles. Figure 5 shows the borehole temperature profiles at AM01–AM06 grouped as with or without basal marine ice.


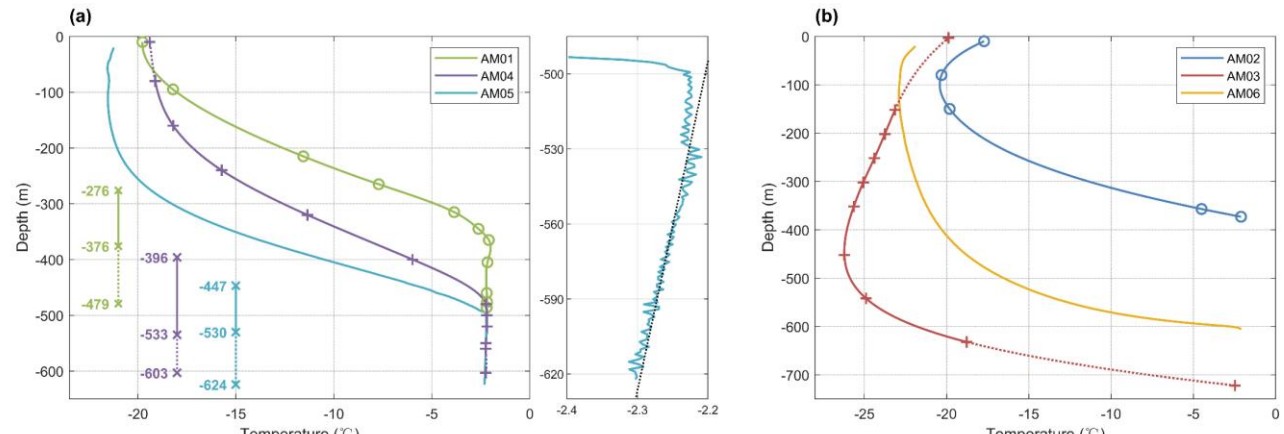

**Figure 5: Borehole temperature profiles (a) at AM01, AM04 and AM05, and (b) at AM02, AM03 and AM06. The subplot of (a) is
the near-isothermal section of AM05, fitted with an in situ seawater freezing line for a salinity of 34.4 psu. Markers (+ and o) indicate
the thermistor points. The depth ranges (lower left of (a)) mark impermeable (solid line), permeable (dotted line) marine ice layers
(Table 3). The temperature of upper and lower surfaces at AM03 and AM04 are inferred from available AWS data and surface air
temperature field (Comiso, 2000), represented by dotted lines.**

AM01, AM04 and AM05, approximately located on the same flowline in the marine ice band (the JP flowline; Fig. 1), show
similar profile patterns (Fig. 5a). The temperature profiles at these sites show nearly isothermal basal layers up to 120 m thick,
which are closely related to the in situ pressure-dependent freezing point of seawater. The correlation is well reflected in the
DTS profile at AM05 (subplot of Fig.5a) where the profile for 530 m–624 m depth is well approximated by a constant salinity
in situ pressure freezing line. The observed pressure-dependent temperature suggests that the marine ice below 530 m depth
maintains a hydraulic connection with ocean below. There is a slight step at 530 m depth, which corresponds to a level above
which fresher water from the drilling process was observed in the borehole. The profile above 530 m depth no longer matches
the pressure freezing line, implying the termination of the hydraulic connection, and at 500m there is an abrupt change in the
temperature gradient. Therefore, we estimate that at AM05, the interface between permeable and impermeable marine ice
(corresponding to the hydraulic connection depths observed at AM01 and AM04) is around 530 m depth (marked in Fig. 5a).
The temperature at the interface between meteoric ice and impermeable marine ice drops from ~-6.2 °C at AM04 to ~-6.8 °C





at AM01 over the period of 102.6 years. If it is assumed that the interface temperature drops at the same rate from AM05 to
AM04, the temperature of meteoric–marine ice interface at AM05 can be estimated to be -5.8 ℃. Combined with the observed
temperature profile of AM05, the interface depth can then be estimated as 447 m (marked in Fig. 5a), corresponding to a
marine ice thickness of 177 m at AM05. This estimation is in good agreement with the marine ice thickness expected on the
basis of vertical strain thinning and basal ice accumulation from AM05 to AM04. The internal temperature gradually increases
along the JP flowline, while the internal temperature gradients remain stable and relatively high (Table 5) above the isothermal
zone at these three sites. At AM05, the upper 200 m of the meteoric ice is nearly isothermal, and the corresponding layer is
thinned to ~100 m at AM04, then approximately dissipated at AM01.

**Table 5: Temperature gradients and derived corresponding heat fluxes at the base of the meteoric ice at the six borehole sites. The**
**heat flux is calculated according to Fourier's law, including the fact that thermal conductivity of ice, $k_{ice}$, is a function of ice**
**temperature as shown in Table 4.**

| Borehole sites | Temperature gradient at the base of meteoric ice (℃ m$^{-1}$) | Corresponding heat flux (W m$^{-2}$) |
|---|---|---|
| AM01 | -0.08±0.01 | -0.18±0.02 |
| AM02 | -0.15±0.01 | -0.32±0.02 |
| AM03 | -0.26±0.03 | -0.54±0.06 |
| AM04 | -0.07±0.01 | -0.15±0.02 |
| AM05 | -0.10±0.01 | -0.22±0.02 |
| AM06 | -0.36±0.01 | -0.75±0.02 |

AM02, AM03 and AM06, in the area without marine ice, show large temperature gradients within 100 m of the lower surface
layer (Fig. 5b). AM02 and AM03 reveal significantly colder ice in the interior of the ice shelf column than at the upper surface,
hereafter referred to as "cold cores" in the ice shelf. The temperature gradients at the base of the meteoric ice, as well as the
corresponding heat flux, derived from the borehole temperature profiles are shown in Table 5. At sites AM02, AM03 and
AM06 without a marine ice layer, the heat flux represents that across the ice shelf/ocean interface.

**3.2 Results from 3-D steady-state temperature simulations**

A series of 3-D temperature distributions are estimated from the simulations with four different basal mass balance datasets
described in Sec.2.2 (Table 2). The simulated temperature profiles at the six borehole sites are obtained from the 3-D
temperature fields (Fig. 6). Normalised depth, $\overline{D}= D/H$ (where $D$ is the depth below the upper surface and $H$ is the ice
thickness), is used to facilitate comparisons between the simulated and measured vertical temperature profiles. This is due to
thickness differences between borehole measurements and the BedMachine data used in the model (Fig. 6). Within the marine





ice band, the total ice thickness at sites AM01, AM04 and AM05 in the model is ~60 m less than the borehole measurements, while the differences at sites AM02, AM03 and AM06 are significantly smaller, no more than 20 m.

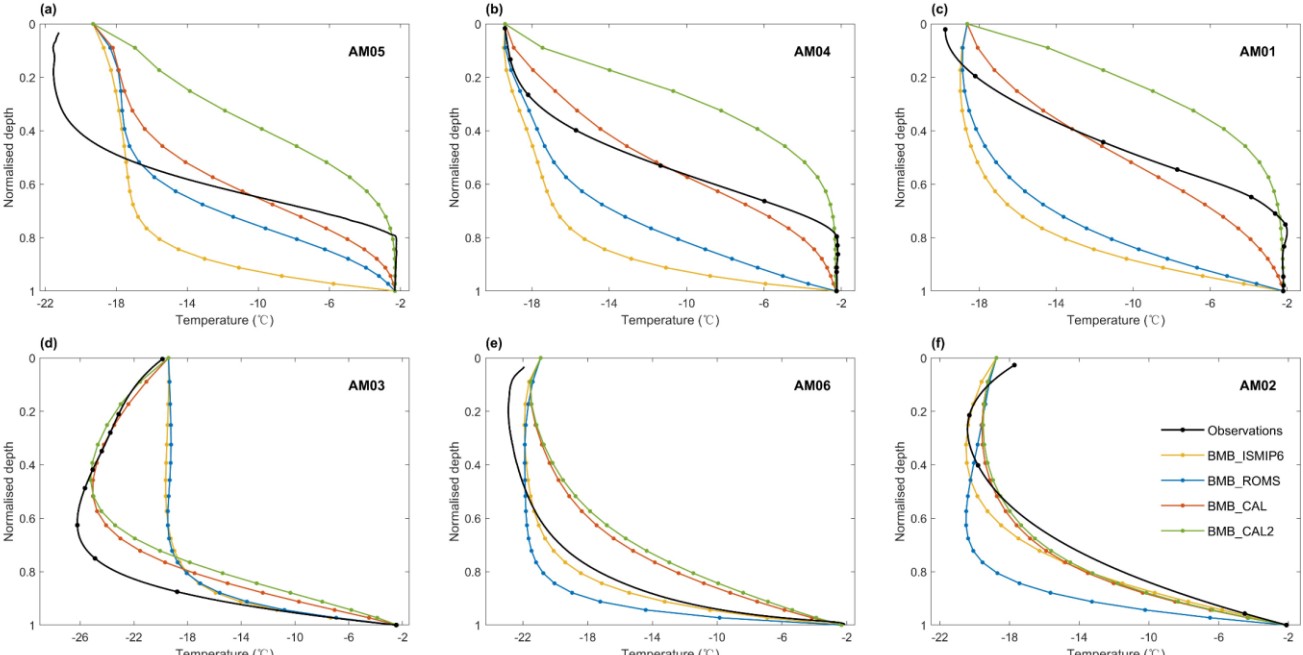

**Figure 6: Normalised comparison between measured and simulated temperature profiles at six borehole sites, shown in Fig.1. AM05, AM04 and AM01 (a, b, c) are on the JP flowline from upstream to downstream with marine ice. AM03, AM06 and AM02 (d, e, f)**
**are from upstream to downstream, experiencing basal melting. Observed temperature profiles are derived from the borehole thermistor (AM01–AM04) and DTS data (AM05, AM06). The 3-D model has 20 layers in the vertical, with circles indicating nodes.**

The extracted simulated temperature profiles for different basal mass balance choices show different patterns at the six sites (Fig. 6), which reflect significant differences between simulated temperature fields. Differences between the simulations and
the borehole measurements in the upper surface are found at each site, especially for AM05, because the upper surface temperature in the simulations is fixed by the Antarctic surface temperature dataset (Comiso 2000). As shown in Fig. 6, the simulation with BMB_ISMIP6 provides reasonable fitting at AM02 and AM06 (within the basal-melt area), but poor fitting at the sites with basal marine ice (AM01, AM04 and AM05) compared with other experiments, which is expected since BMB_ISMIP6 purely represents basal melting in those marine ice regions (Fig. 3a). The simulation with BMB_ROMS fits
the borehole temperature profiles slightly better at the sites with marine ice, since basal accretion is considered in BMB_ROMS, while at AM02 and AM06 the effect of higher melt rates (see Fig. 3b) leads to poorer agreement than for BMB_ISMIP6. The BMB_ISMP6 and BMB_ROMS simulations give very similar but poor matches at AM03. The simulation with BMB_CAL provides better fitting results at most of the borehole sites, but still does not reconstruct the near-isothermal marine ice layer



at the bottom. In contrast, the simulation with BMB_CAL2 shows a much thicker marine ice layer, which is visually close to

the pressure freezing temperature line in the permeable marine ice layer. However, the manually increased basal accretion in

BMB_CAL2 also leads to a severe overestimation of temperatures for the colder ice above the near-isothermal layer. For the

sites outside the marine ice band (AM01, AM03 and AM06), the simulated temperature profiles from BMB_CAL and

BMB_CAL2 show little difference. In general, the temperature field simulation with BMB_CAL best fits most of the borehole

temperature profiles visually, which suggests that BMB_CAL is more representative of the real mass balance at the ice–ocean

interface. Therefore, we mainly conduct detailed analysis and discussion on the simulations with BMB_CAL in the remainder

of the paper.

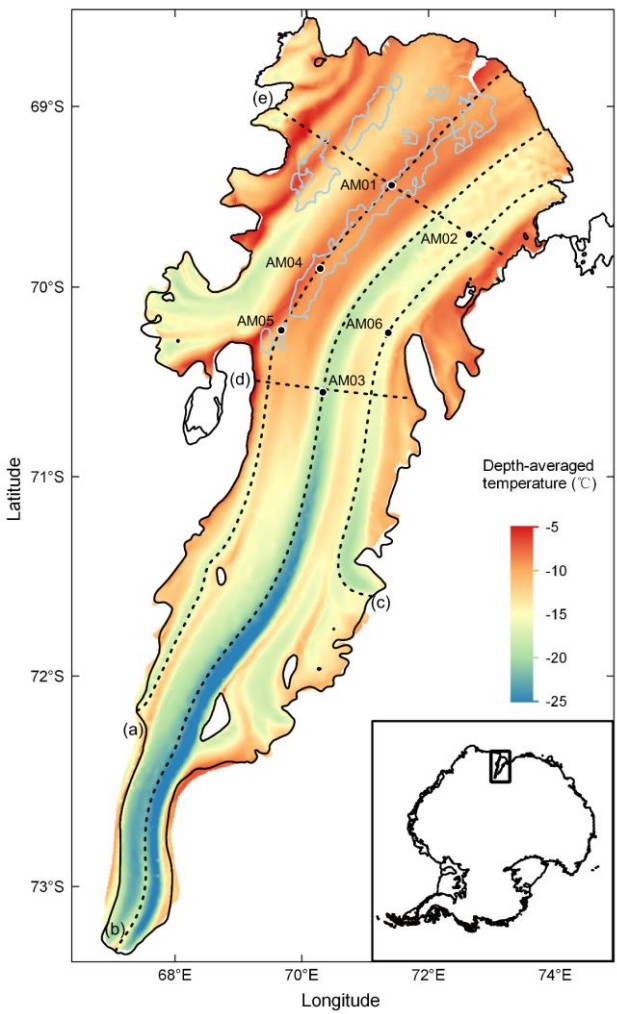

**Figure 7. Depth-averaged temperature distribution of the AIS from the 3-D simulations with BMB_CAL. Three flowlines, derived**

**from simulated velocity field, and two crossing lines are marked with dash lines. Marked letters (a–e) correspond to the sequence of**





**cross-sections in Fig. 8. Marine ice band with estimated thickness greater than 100 m is shown with the light blue contours (Fricker et al. 2001). Inset shows the location of the Amery Ice Shelf in East Antarctica.**





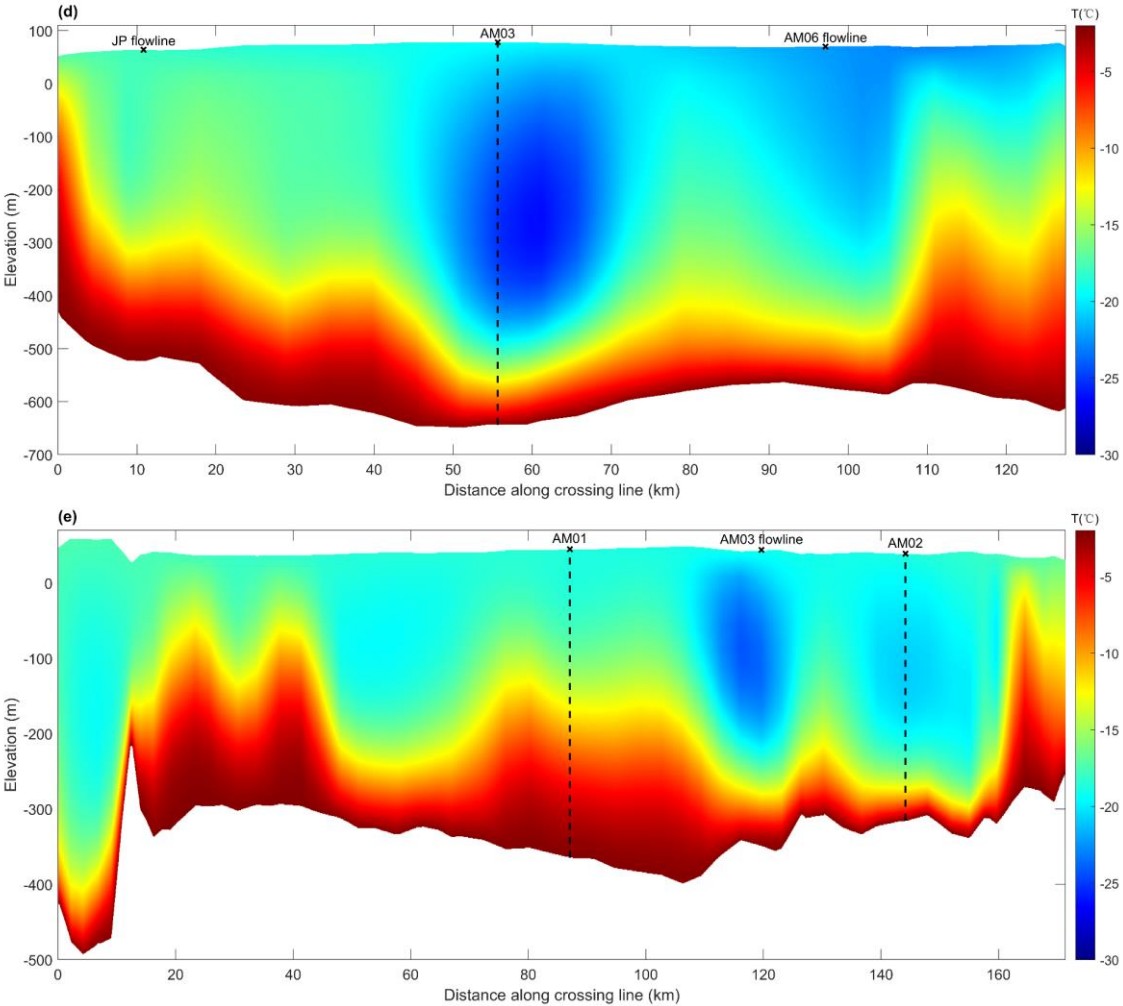

Figure 8: Simulated temperature sections along (a–c) and across (d–e) the three flowlines from the 3-D temperature simulations with BMB_CAL, and their location is shown in Fig. 7. The three flowlines are (a) JP flowline, (b) AM03 flowline and (c) AM06 flowline from west to east derived from the simulated velocity field. The positions of key points on the flowline are marked; all boreholes are shown with vertical dash lines. In coordinates, "distance along flowline" is relative to the grounding line. Inset presents the basal mass balance (BMB_CAL) along the three flowlines with key points marked.

To visualize the simulated steady-state temperature field, the depth-averaged temperature distribution and a series of temperature sections along and across the three flowlines are extracted from the simulation of BMB_CAL, as shown in Fig. 7 and Fig. 8. We note that the three flowlines here are derived from the simulated velocity field of inversion simulations, which is almost the same as those obtained from the MEaSUREs Antarctica Ice Velocity Map (Rignot et al. 2017). The distribution pattern of the depth-averaged temperature is strongly aligned with the flowlines (Fig. 7). The average ice temperature gradually increases along the flowlines downstream, which is the clearest along the AM03 flowline.



All simulated temperature sections (Fig. 8) show internal cold ice advected from upstream tributary glaciers, originally formed due to downward advection of ice from cold, high-elevation regions far inland. The minimum internal temperature on the

AM03 flowline is ~-28 °C, lower than that of the JP flowline (~-22 °C) and AM06 flowline (~-24 °C). The cold ice is gradually warmed along flowlines by heat conduction from warmer ice both above and below as it propagates toward the ice front. Along the flowlines, the tens of kilometres near the grounding line reflect the transition of the thermal structure from grounded ice to floating ice shelf, all of which experience the steepening of the basal temperature gradient, associated with active basal melting. Along the JP flowline (Fig. 8a), the internal cold core, warmer than that of the AM03 flowline, gradually warms up and

dissipates around the JP (Fig. 8a, d). Marine ice starts to accrete around 150 km downstream of the grounding line (see inset in Fig. 8a), where basal temperature gradient starts to decrease. The continuous marine ice accretion is maintained downstream of AM05, corresponding to the substantially consistent temperature regime. Along the AM03 flowline (Fig. 8b), the ice thickness decreases rapidly within 100 km downstream of the grounding line, accompanied by a significant steepening of the temperature gradient of the lower part of the ice shelf, where it experiences considerable basal melting up to 20 m a⁻¹. The

large basal temperature gradient gradually eases until 200 km downstream of the grounding line where a basal mass balance close to zero is reached (inset in Fig. 8b). The cold core, approximately 30 km wide at AM03 (Fig. 8d), is mainly composed of the cold continental ice from the Mellor and Lambert Glaciers, flowing through the southern grounding line of the AIS. Along the AM06 flowline (Fig. 8c), the temperature section illustrates the formation process of cold core ice. Internal temperature of the ice shelf upstream of AM06 is close to the surface temperature (Fig. 8c, d). The surface temperature

increases significantly downstream of AM06, at which time the internal temperature is lower in comparison, resulting in the formation of a cold core. It can be found at AM02 (Fig. 8c, e) and propagates downstream all the way to the ice front. The basal temperature gradient along the AM06 flowline is relatively small upstream of AM06, associated with a gentler basal melting than the other two flowlines (inset in Fig. 8c).

The transverse temperature sections exhibit a great variation of the thermal structure across the ice flow (Fig. 8d, e). Based on the AM01–AM02 transverse temperature section (Fig. 8e), in addition to the cold core on the AM03 flowline and at AM02, there is also internal cold ice from Scylla and Charybdis Glacier to the west (50–70 km along the transverse section). The basal warm ice layer at AM01 is ~30 km wide (75–105 km along the transverse), which correlates well with the pattern of the marine ice band (Fig. 7). Similarly, the basal warm layer to the west, ~30 km wide (15–45 km along the transverse), also corresponds

to the distribution of the western marine ice band (Fig. 7).

Based on the simulated steady-state temperature field of BMB_CAL, the distribution across the AIS of the depth-averaged ice stiffness factor, $B(T_h)$, is calculated. The distribution pattern of the depth-averaged flow rate factor is very similar to that of the depth-averaged temperature. The detailed calculation process and the distribution are shown in appendix A. The flow rate

factor in most areas of the AIS (86% in area) is between $1 \times 10^8$ and $1.8 \times 10^8$ Pa s$^{1/3}$, with an average value of $1.48 \times 10^8$ Pa s$^{1/3}$.





The factor achieves the maximum in the southern grounding zone, where the ice is the coldest, and decreases downstream associated with progressive warming.

### 3.3 Results from the 1-D temperature column simulations

The use of the column model allows for the horizontal advection of cold ice when forced with the continuous evolution of
thermal structure with time-evolving boundary conditions. A series of column simulations provide solutions of the 1-D advection-diffusion equation with specific boundary conditions. Figure 9 compares the vertical temperature profiles from the 1-D simulations, 3-D simulation of BMB_CAL and observations. The column simulations achieve slightly better fitting results than the 3-D simulations at AM01, AM03 and AM05. From AM05 to AM04, the prescribed surface temperature is warmer by 1.8 °C (see Table 3) and is reflected in the modelled cold core within 200 m of the upper surface in AM04 temperature profiles
(Fig. 9b). The 1-D simulated temperature profiles have higher vertical resolution in the vertical direction than that from the 3-D simulations.

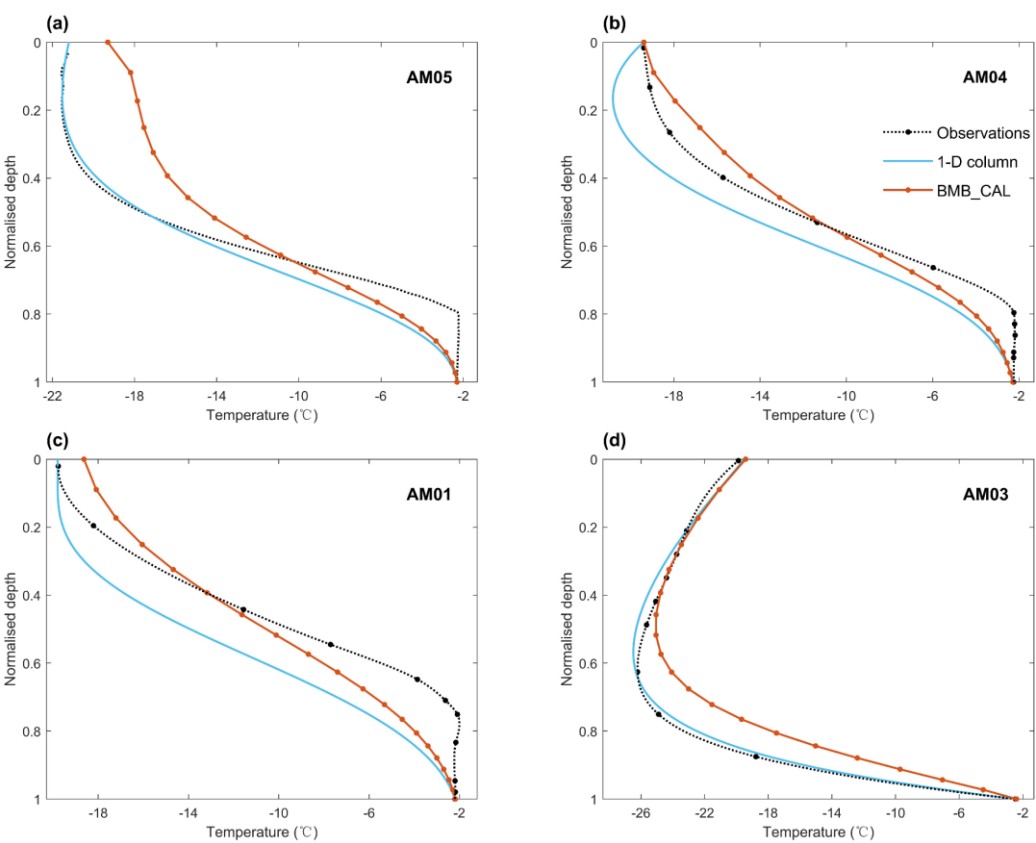

**Figure 9: Normalized comparisons of temperature profiles from borehole observations, 1-D column simulations and 3-D simulations.**
**AM05, AM04 and AM01 are on the JP flowline from upstream to downstream; AM03 is on the AM03 flowline. The 3-D model has 20 layers in the vertical, with circles indicating nodes. There are 100 layers in the column model, so individual nodes are not marked.**



## 4 Discussion

### 4.1 Factors determining the thermal structure and its spatial pattern

Distinct thermal structures are evident for the areas with or without a basal marine ice layer from the borehole temperature
profiles and the simulations. Vertical advection, determined by surface and basal mass balance (melting and freezing) and
vertical strain, strongly affects the vertical thermal regime at each location. Thermal conduction is also significant in the
vertical direction, smoothing the temperature profile. Horizontal advection transports the local thermal regime from one
location to another, thereby establishing the spatial pattern of the temperature distribution. For the marine ice layer, its distinct
material properties and the hydraulic interaction between the porous layer and the ocean below dominate the local thermal
regime.

Focussing on vertical advection, basal melting creates downward ice advection deep in the ice shelf. The internal ice, with
lower temperature, is therefore advected closer to the base where the ice is warmer due to the ocean contact, resulting in a
significant increase in temperature gradient near the base of the ice shelf. This effect can be seen in both borehole temperature
profiles and the simulations. Large basal temperature gradients are observed at AM02, AM03 and AM06 sites in the basal
ablation zone (Fig. 5b; Table 5). In the simulated temperature field for BMB_CAL, the maximum basal temperature gradient
of -0.8 °C m$^{-1}$, occurs in the southern part of the AIS, where basal melting is ~7 m a$^{-1}$ and downstream of a region where melt
rate exceeds 15 m/a. Large gradients have also been observed at the base of other Antarctic ice shelves, such as at the S1 site
on the Fimbulisen (Orheim et al 1990a,b; modelled in Humbert 2010) and the McMurdo Ice Shelf (Kobs et al. 2014). For
McMurdo Ice Shelf, the average observed temperature gradient at base of the borehole is -0.38 °C m$^{-1}$, associated with an
estimated basal melt rate of 1.05 m a$^{-1}$. In contrast, basal refreezing creates an upward advection of warm accreted marine ice
and decreases the basal temperature gradients, a feature illustrated by the simulated temperature profiles at AM05, AM04 and
AM01 within the marine ice band (Fig. 6a, b, c; Fig 8a). In the temperature section along the JP flowline (Fig. 8a), the
dissipation of the cold core at JP is associated with the upward advection of basal warm ice. Accretion of marine ice dominates
the thermal regime of the lower part of the ice shelf downstream of JP. Similarly, the surface mass balance
(accumulation/ablation) also causes vertical advection of temperature and affects the pattern of the vertical thermal regime.
However, the surface mass balance is always smaller in magnitude than the basal mass balance and has less spatial variation
across the AIS. According to the RACMO model data (Melchior Van Wessem et al. 2018), the surface accumulation rates of
the AIS averaged from 1979 to 2017 range from approximately -0.03 to +0.6 m a$^{-1}$ ice equivalent. In contrast, the range of the
basal mass balance is approximately -30 to +3 m a$^{-1}$ (Adusumilli et al. 2020). Considering the magnitude of the two mass
balances, the effect of surface mass balance on thermal structure should be significantly less than that of the basal mass balance.
We note that the nearly isothermal profile of the upper meteoric ice observed at AM05 (Fig. 5a) cannot be explained by general
surface accumulation, and that horizontal ice advection is the more significant ingredient as discussed below. Vertical strain
also contributes to the vertical temperature profile. Vertical strain results from horizontal divergence in the depth-averaged





flow regime, and in ice shelves is mainly due to extensional flow. As ice flows towards the calving front, it generally accelerates and thins. Strain thinning acts to compress the ice column and steepen the vertical temperature gradient in the ice shelf (Craven et al. 2009). Comparing the borehole temperature profiles at AM01 and AM04 (Fig. 5a, and Table 5) above the marine ice layer, the internal temperature gradient at AM01 is significantly greater than at AM04, which reveals the effect of strain thinning.


Horizontal ice advection transports cold ice originally deposited at high-elevation on the grounded ice sheet to the downstream ice shelf and gives rise to the formation of the core of cold ice, well reflected in the temperature sections from some of the 3-D simulations (e.g., BMB_CAL in Figure 8) and observed at AM03. Along the AM03 and AM06 flowlines (Fig. 8b, c), the cold continental ice from the upstream tributaries persists to the ice front, and dominates the internal temperature regime along

the ice shelf. Compared with the cold core observed at AM02 (Fig. 5b), that of AM03 is proportionally much lower in the vertical column, which is determined by the origin of the coldest ice, and the influences of surface accumulation and basal melting, as well as the evolution of ice surface temperature. Our results indicate that the evolution of the thermal structure along the flowline is accompanied by the warming of the meteoric ice, contributed by internal thermal conduction.

The porous structure of the lower part of the marine ice and its hydraulic connection with ocean below give rise to the near-isothermal basal layer (Fig. 5a). Craven et al. (2009) regarded the hydraulic connection depth encountered in drilling as an approximation to the effective pore close-off depth. Beneath the hydraulic connection depth, the permeable marine ice has interconnected channels and cells, filled with seawater (Craven et al. 2009). The relatively free movement of the seawater within the pores keeps the ice–seawater mixture at the in situ pressure-dependent seawater freezing temperature (McDougall

et al. 2014), as shown in the subplot of Fig. 5a. Above the hydraulic connection depth, there is still apparently residual brine trapped in the pores of the impermeable marine ice, observed through borehole video imagery and in ice core samples (Craven et al., 2005, 2009). These brine inclusions decrease in volume by freezing at the walls, becoming saltier and lowering the freezing point of the residual brine, allowing the two-phase material to further cool. Available salinity measurements on ice core samples recovered from the AM01 borehole reveal that the total salinity of the upper impermeable layer is very low

(Craven et al. 2009). Above the permeable layer, the consolidated marine ice gradually cools, which is confirmed by the temperature drop of the meteoric-marine ice interface from AM04 to AM01 (Craven et al. 2009).

**4.2 Implications of the AIS thermal structure simulations**

Both the 1-D and 3-D simulations produce imperfect fits to the observed vertical temperature profiles at the six borehole sites (Fig. 6; Fig. 9). The discrepancies are most notable in the regions that have a marine ice layer. Our modelling approach contains

assumptions, limitations and sensitivities that are pertinent to consider when interpreting the model outputs, some of which may contribute to this model-data discrepancy. We discuss these limitations and some possible avenues to address them before proceeding to the implications of our present modelling studies.



### 4.2.1 Limitations and avenues for model improvement

Our modelling approach treats marine ice and meteoric ice the same way. The presence of seawater in the thick porous or
permeable layer of marine ice is almost certainly the main cause of the discrepancy between the shape and gradient of modelled
and observed borehole temperature profiles where marine ice is present (Fig. 6 a, b, c). This limitation applies to both our 1D
and 3D simulations. A more sophisticated treatment is required to capture the thermodynamics and evolution of the two-phase
porous seawater saturated marine ice layer.

Similarly, the detailed interactions between the porous firn layer and the atmosphere are not incorporated in our simulations,
with instead only a Dirichlet temperature condition at the upper surface of solid ice. The model, therefore, responds less rapidly
to the atmospheric temperature changes than the real system. Again, this would affect both 1D and 3D simulations. In
combination with the choice of surface temperature forcing it might explain discrepancies in the formation of a subsurface
cold ice band at AM02 (Figure 6f). The formation of the subsurface cold-core at AM04 only in the 1-D simulations (Fig. 9b)
presumably arises from the different choice of surface forcing, which reflects a decision to make an independent simulation
more closely connected to the borehole observations, rather than performing a model comparison. The surface and bottom
temperatures used to force the column model are linearly interpolated between key time stamps, which may also introduce
some inaccuracies.

The 3-D temperature simulations make the steady-state assumption, which neglects the impact of long-term (e.g., decadal to
millennial-scale) changes and seasonal signals in boundary conditions. Considering that it takes ~1100 years for ice to reach
the ice front from the southern grounding zone, this could cause a bias in our simulated temperature distribution. Indeed, given
the century time scales for ice to travel between the JP flowline boreholes, surface temperature changes over the 20[th] Century
could have some role. In particular, if the ice sheet is in steady state then the emergence velocity at the upper surface should
be equal and opposite to the surface mass balance, which is not the case in our simulations. We find large magnitude and strong
spatially variations in the emergence velocity, with a few areas greatly exceeding ±5 m/a. While such patterns could plausibly
exist in the real system as a manifestation of advecting geometric features, their impact on our steady-state temperature
simulations is to impose a non-physical advection of ice through the upper surface. The impacts of the steady-state assumption
for simulating temperature are discussed further by Gladstone et al. (2021, in preparation). The strongest impacts of these
emergence velocity features are on the grounded ice sheet basal temperatures (which feature a cold bias), and also, to some
extent, on the vertical temperature profile at the grounding line, which is highly relevant to our 3-D model outputs through
downstream advection into the shelf. The study by Gladstone et al (2021, in preparation) found that the dynamic upper surface
boundary condition introduced in Section 2.2.2 (Eq. 5) reduced these cold bias effects.



The nature of the forcing we impose for our 1-D transient temperature simulations is also equivalent to the steady state assumption: the evolution of column forcing as a column is advected from inland ice sheet to ice front is based on present day conditions, though a column currently at the ice front would have crossed the grounding line several hundred years ago.

The 3D inversion and steady-state simulations use spatial observational datasets for ice geometry and velocity, thermal forcing
etc., which are assumed to be temporally consistent as well as being constant over time. In practice, these data have been gathered over different time intervals, and possible temporal inconsistencies, as well as actual errors in the data, could lead to errors in the optimised model state. However, given that the LAGS, unlike some other Antarctic catchments, is not currently changing rapidly over recent decades (King et al., 2007; Pittard et al., 2015; Yu et al., 2010), lack of data synchronicity is not likely to be a major issue.


The Antarctic bedrock is difficult to observe with spatially consistent accuracy, and even ice shelf thickness is not well constrained because it is often derived assuming local hydrostatic equilibrium (particularly where marine ice accretion prevents radar measurements) which means that very high accuracy in upper surface elevation is required in order to infer draft, and that uncertainties about ice density profiles remain. As mentioned, the thickness discrepancies between the six borehole
measurements and the BedMachine data used in the modelling reach a maximum of 70 m (AM01), with larger discrepancies at the marine ice locations. Given the extensive available observations of the upper surface velocities (used here as an optimisation target for the horizontal velocities), the simulated 3D flow regime is very strongly dependent on the geometry of the lower surface elevation, and highly sensitive to uncertainties in lower surface elevation. It is also possible that the noisy emergence velocity mentioned above could be attributable to lower surface elevation errors rather than the steady state
assumption. On the ice shelf the basal topography combines the dynamical ice shelf basal boundary condition, with forcing prescribed by the BMB as major influence on the vertical aspect of the 3D flow.

In summary, while there are sources of uncertainty that can affect the flow dynamics and hence details of the vertical advection, it is clear that the dynamical boundary condition at the ice shelf base (Section 2.2.2) produces markedly different temperature
profiles for the various basal mass balance forcing choices, and that (as anticipated) the inadequate treatment of thermodynamics of the porous marine ice layer leads to less success in simulating the temperature profiles in regions where marine ice is present. In order to quantify the relative importance of model limitations, several further studies would be informative. Incorporating the upper surface mass balance (and possibly also the lower surface mass balance) into the cost function during the inversion might provide a less noisy emergence velocity, allowing quantification of its impact. Feeding
back the newly simulated steady state temperature fields into further inversions for the parameters $\beta$ and $E_\eta^2$ would help to estimate the net effect of choosing between a long timescale spun up temperature field from a mechanistically simpler ice sheet model and a steady state assumption within a more sophisticated model setup. Ice shelf only simulations in which alternative



temperature distributions at the grounding line are imposed would also help to assess the impact of the grounded ice thermal regime on the ice shelf thermal regime.


However, the most significant shortcoming of the present modelling clearly concerns the representation of marine ice, particularly the porous lower layer. The two-phase character of the permeable marine ice at in situ seawater freezing temperature is not represented and requires a more sophisticated thermodynamic treatment, including the processes of consolidation at the point of pore closure; the evolution from the initial deposition of frazil ice platelets (Galton-Fenzi et al 2012) and the hydraulic interaction with the underlying ocean. Just as the thermodynamics of ice sheet models was extended to treat temperate ice with a freshwater content (e.g., Greve, 1997; Aschwanden et al., 2012; Schoof and Hewitt, 2016), further developments are required for marine ice, and the basal boundary conditions will also involve porosity as well as ice accretion rates. Fortunately, these topics are already being explored in situations ranging from sea-ice (including sub-ice platelet layers) to the ice-ocean interfaces in icy moons of the outer solar system (e.g., Buffo et al 2018; Buffo et al., 2021). The current study aims primarily to assess the presented simulations in comparison to observations, and these suggested modelling developments are noted here for relevance to future studies.

### 4.2.2 Implications of the simulations

The differences between the simulated temperature profiles for four different basal mass balance datasets (Fig. 6) demonstrate a high sensitivity of the ice shelf thermal structure to the pattern of basal melting and freezing. Melting leads to downward advection of ice and a steeper basal temperature gradient. Freezing accretes warm ice at the base and a lower basal temperature gradient. The simulations show that if this is simple consolidated ice then, except for very high accretion rates, that basal gradient quickly increases in the interior of the shelf as the heat is also conducted upwards. The presence of the porous marine ice layer modifies that simple picture, until the marine ice has been consolidated (as discussed by Craven et al 2009). The 3-D simulations using BMB_CAL as forcing in the basal Dirichlet condition on ice velocity provide the best fit (assessed through visual inspection of Fig. 6) to the borehole temperature observations, indicating that this dataset is closer to the actual mass balance at the ice–ocean interface. Adusumilli et al. (2020) also estimated the thickness of marine ice that would be accreted under the AIS from the BMB_CAL data, with good agreement to the reported borehole measurements. Given the differing magnitudes and spatial patterns of basal accretion shown in Figure 3, it is not surprising that the comparisons between simulated and measured temperature profiles in marine ice locations (Fig. 6a, b, c). show marked differences between the four simulations. The BMB_ROMS profiles are far from isothermal for the basal marine ice layer. Furthermore, in the progression from AM05 to AM01 they increasingly depart from similarity to the BMB_CAL profiles, tending towards those for BMB_ISMIP6 which involves no basal accretion at all. The 3-D simulations of BMB_CAL2 produce a significant nearly isothermal basal layer (Fig. 6a, b, c), but this is likely just due to a very high accretion rate of marine ice, as the model lacks the physics to simulate the thermal regime of the permeable marine ice which lies on the in situ pressure freezing line. The temperature gradient at the top of the marine ice band is unable to represent the sharp change seen in the borehole measurements,





and the upward advection imposed by this high basal accretion in BMB_CAL2 leads to serious discrepancies in the temperature profiles in the upper part of the ice shelf.

In general, the thermal structure of ice shelves influences ice rheology and therefore also the dynamical regime (Humbert 2010,
Budd & Jacka 1989). For ice shelf flow, which is principally governed by stresses acting in the horizontal plane, the depth-averaged effect of the ice stiffness factor, $B(T_h)$, quantifies the dependence of ice viscosity on temperature, as discussed in previous studies (e.g., Humbert 2010, Craven et al. 2009). Our depth-averaged ice stiffness factor for the AIS is similar to that of the Fimbulisen in magnitude and distribution pattern, in which the value of the factor decreases downstream associated with warming process (Humbert, 2010). The 3-D field of the ice stiffness factor calculated from the current temperature simulations
can be used for future ice dynamic modelling studies of the AIS. Even with the deficiencies of the temperature simulations in the marine ice zones, our depth-averaged stiffness factor already shows the important effect of temperature structure. However, the modelling approach in the current study also treats deformation of marine ice in the same way as meteoric ice. There is very limited experimental data about the deformability of marine ice. While Dierckx and Tison (2013) found that consolidated marine ice deformed similarly to meteoric ice at the same temperature, they did not explore the tertiary flow regime, where
the influence of impurities on dynamic recrystallization might be significant. It also seems unlikely the permeable layer would deform like meteoric ice, so that our current depth-averaged ice stiffness factor is likely an overestimate for regions where the marine ice thickness is a significant fraction of the whole.

The thermal structure of the AIS shows strong dependence on that of the upstream tributary glaciers. The history of the cold
cores along flowlines (Fig. 8) shows that the thermal structure of the grounded ice sheet is imposed on the downstream ice shelf. The biggest cold core of the AIS, approximately 30 km wide at AM03 (Fig. 8d), is composed of ice from the Mellor and Lambert Glaciers, which supply most of the ice at the southern grounding line. Elsewhere in Antarctica, a similar cold core is also detected in the Fimbulisen, originating from the major inflowing ice stream Jutulstraumen (Humbert, 2010), and such cold cores may be expected as common features in Antarctic ice shelves, especially where fast flowing ice streams are present
to advect the cold ice through the shelf. Due to the formation of the internal cold core of ice far inland, with long timescales for advection of this cold ice into the shelf, its structure in the AIS is unlikely to be affected by climate changes on decadal timescales. Recent studies also suggest that the AIS is, and will continue to be, stable (e.g., Pittard et al., 2017). However, the porous marine ice layer could respond more rapidly to any changes in ocean circulation below through hydraulic interactions (Herraiz-Borreguero et al., 2013).

**5 Conclusion**

The thermal structure of the Amery Ice Shelf and its spatial pattern are evaluated and analysed through borehole observations, 3-D steady-state temperature simulations and 1-D temperature column simulations. We present vertical temperature profiles





of the Amery Ice Shelf at six borehole sites, AM01–AM06, based on thermistor and DTS measurements, indicating distinct thermal structures along flowlines in regions with and without marine ice. In AM01, AM04 and AM05 boreholes that have a

permeable basal layer of porous marine ice, approximately 100 m thick is present, which appears to conform to the pressure-dependent seawater freezing temperature. In AM02, AM03 and AM05 boreholes that are experiencing active melting, large temperature gradients up to -0.36 °C m$^{-1}$ are found at the base. An interior core that is colder than both the surface and basal ice, having been advected from cold, high elevations in the ice sheet by the major tributary glaciers is found at AM03. The 3-D simulations produce a set of 3-D steady-state temperature fields for four different basal mass balance datasets, and the

differences between them demonstrate a high sensitivity of the thermal structure to the pattern of basal melting and freezing. Based on the comparisons with borehole observations, the 3-D simulation with BMB_CAL (Adusumilli et al., 2020) is considered to best approximate the real thermal structure of the AIS, which indicates that BMB_CAL is more representative of the real basal mass balance. The simulated temperature field shows a great variation of the thermal structure across the ice flow and illustrates spatial evolution of the AIS thermal structure, dominated by the progressive downstream warming of the

cold cores of ice from the tributary glaciers. The depth-averaged temperature-dependent ice stiffness factor, $B(T_h)$, of the AIS is also calculated from the BMB_CAL temperature field to quantify the dependence of ice viscosity on temperature and demonstrate its influence on dynamics. The 3-D field of $B(T_h)$ can be used for further dynamic modelling studies of the AIS. The 1-D simulations, based on time-stepping to follow columns of ice along two flowlines, with corresponding time-stepping of the column boundary conditions further exhibit the formation of the thermal structure. They provide simulated temperature

profiles along the flowlines, in slightly better agreement with the borehole observations than the 3-D simulations.

Our results illustrate that vertical advection, determined by basal and surface mass balance as well as vertical strain, strongly affects the vertical thermal regime at each location. Horizontal advection transfers these effects downstream along with the ice flow, cumulatively establishing the spatial pattern of the temperature distribution. For the marine ice layer, its porosity and

interactions with ocean below determine the local thermal regime, which cannot be reproduced in the current simulations. Based on our results and the related thermal analysis of the Fimbulisen (Humbert, 2010), we expect that similar thermal structures dominated by cold core ice may commonly exist among the Antarctic ice shelves, especially where thick fast-moving glaciers feed into the ice shelf. Given the millennial timescales of the evolution of the AIS thermal structure, it is unlikely to be affected by climate changes on decadal timescales. However, the porous marine ice layer is likely to be susceptible to

potential changes in basal mass balance and ocean circulation through hydraulic interactions (Herraiz-Borreguero et al., 2013).

This study presents the first quantitative analysis of the 3-D temperature field of the Amery Ice Shelf. The 3-D and 1-D modelling approach in this study can also be used for thermal analysis of other ice shelves and ice sheets. The discrepancy between observations and model simulations, due to a series of limitations in the 3-D and 1-D models, indicates where

improvements are required to permit better quantification of the thermal structure. In particular, this identifies the need for ice shelf/ocean coupled models with improved thermodynamics for marine ice and more comprehensive evaluation of boundary





conditions. Given the significant influence of ice temperature on the deformability of ice, the simulated steady-state temperature field as well as the processed borehole observations provide a starting point for further studies on the rheology and dynamics of the Amery Ice Shelf.

**Appendix A: Calculation of depth-averaged flow rate factor**

What we term the ice stiffness factor $B(T_h)$, as a function of ice temperature relative to the pressure melting point, is often parameterised by an Arrhenius law form for the ice deformation rate factor (or a pair of matched parameterisations) as

$$B(T_h) = (A_0 e^{-Q/RT_h})^{-1/n} \qquad (A1)$$

where the included physical parameters are given by Paterson (1994), listed in Table A1. The simulated 3-D steady-state

temperature field with BMB_CAL has 20 equally spaced layers in the vertical direction. The factor of each layer is calculated at each horizontal grid, and then the depth average calculation is made. The distribution of the calculated depth-averaged temperature-dependent ice stiffness factor is shown in Fig. A1.

**Table A1: Standard physical parameters for the Arrhenius law (Paterson 1994).**

| Parameters | Symbol | Value | Units |
|---|---|---|---|
| Stress exponent | $n$ | 3 | - |
| Pre-exponential constant | $A_0$ | $3.985 \times 10^{-13}$ (for $T \leq 263.15$ K) <br> $1.916 \times 10^3$ (for $T > 263.15$ K) | $s^{-1}$ $Pa^{-1}$ |
| Activation energy | $Q$ | 60 (for $T \leq 263.15$ K) <br> 139 (for $T > 263.15$ K) | kJ $mol^{-1}$ |
| Universal gas constant | $R$ | 8.314 | J $mol^{-1}$ $K^{-1}$ |






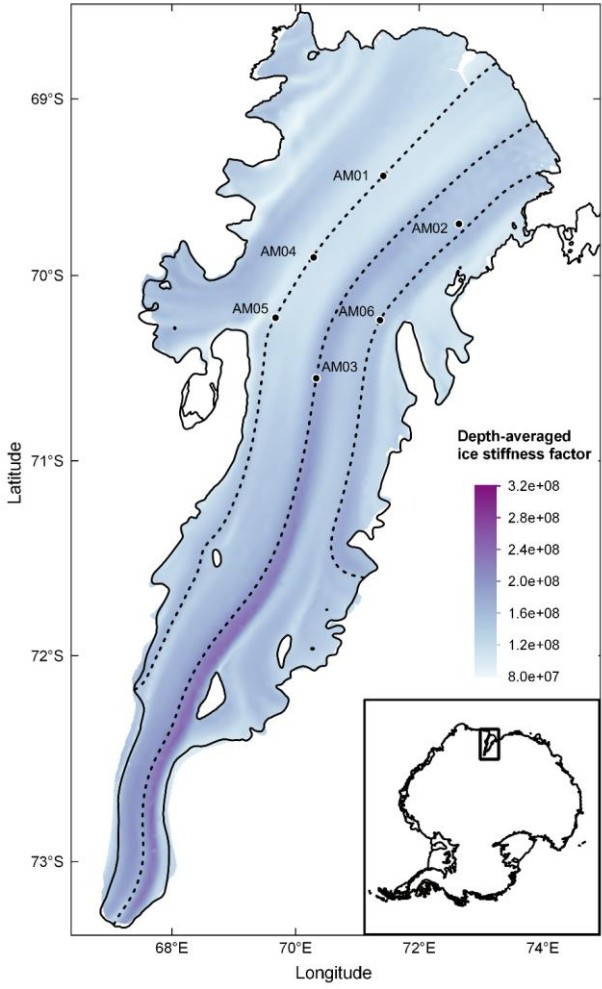

**Figure A1. Depth-averaged temperature-dependent ice stiffness factor $B(T_h)$, in Pas$^{1/3}$, of the AIS derived from simulated temperature filed. Three flowlines, derived from simulated velocity field, are shown with dash lines. The six boreholes are marked and inset shows the location of the Amery Ice Shelf in East Antarctica.**

**Code availability**

The 3-D full Stokes model and 1-D free-surface column model are implemented using Elmer/Ice Version: 8.4 (Rev: d296bb) with the scripts at https://github.com/ElmerCSC/elmerfem.git.
**Data availability**

The borehole internal temperature observations are provided through project AAS 4096 maintained by Australian Antarctic
Data Centre (AADC). The borehole surface temperature data are provided by the Automatic Weather Station (AWS) retrieved
from https://data.aad.gov.au/metadata/records/antarctic_aws, maintained by Australian Antarctic Division (AAD) and AADC.

**Author contribution**

YW, CZ, BG and RG designed the experiments together. RG implemented the 3-D simulations, and YW implemented the 1-
D simulations. YW collated, processed and analysed the thermistor data for AM01–AM04 and RW collated, calibrated and
analysed the DTS data for AM05–AM06. YW drafted the paper. All authors contributed to the refinement of the experiments,
the interpretation of the results and the final manuscript.

**Competing interests**

The authors declare that they have no conflict of interest.

**Acknowledgements**

Yu Wang, Chen Zhao, and Ben Galton-Fenzi are supported under the Australian Antarctic Program Partnership (Project ID
ASCI000002). Roland Warner is a University Associate with the AAPP. The Australian Antarctic Program Partnership is led
by the University of Tasmania, and includes the Australian Antarctic Division, CSIRO Oceans and Atmosphere, Geoscience
Australia, the Bureau of Meteorology, the Tasmanian State Government and Australia's Integrated Marine Observing System.
Rupert Gladstone is supported by Academy of Finland grant number 322430. This project received grant funding from the
Australian Government as part of the Antarctic Science Collaboration Initiative program. We thank Susheel Adusumilli and
colleagues for providing their estimates of marine ice thickness of the Amery Ice Shelf. Lastly, we acknowledge the
contributions of all the AMISOR field teams, operations support staff and the support of Australian Antarctic Division through
projects AAS 1164 and 4096.

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
