# Peer review of "Thermal structure of the Amery Ice Shelf from borehole observations and simulations"

_The Cryosphere, 2021_

## Referee Comment (RC1)

**Review of**
**Thermal structure of the Amery Ice Shelf from borehole observations and simulations by Wang et al.**

Thomas Kleiner

**General comments**

The authors present steady-state temperature-depth profiles through the ice at several locations along two distinct flow-lines at the Amery Ice Shelf, East Antarctica, derived from several years of observation within boreholes using thermistor strings and fibre-optical temperature sensing. They further study the sensitivity of simulated steady-state temperature fields on basal mass balance forcing using the full-Stokes ice sheet model Elmer/Ice in a three-dimensional model domain. Too add some sort of time dependency, the authors also conduct simulations with one-dimensional 'temperature only' simulations, where the boundary conditions vary along the path and vertical strain is prescribed from the borehole observations.

The authors state that the internal temperatures represent a record of the past climate and the thermal conditions upstream, and I would add, that they also serve as a very valuable observable to validate and/or calibrate numerical ice flow models. Especially as ice sheet model assessment is still mainly based on ice sheet wide integral or two-dimensional quantities (Goelzer et al., 2020; Seroussi et al., 2020) ignoring the information that is also preserved in the ice column, e.g. layer depths or temperatures.

The authors find, that basal mass balance forcing derived from remote sensing data leads to the best agreement between the model and the data (given the limitations of the model). I think, this is not surprising as long as ocean models only 'inform' the ice flow model about the basal melt rate without taking the heat flux within the ice towards the ice-ocean interface into account. The same holds for basal melt parametrisations that additionally do not account for basal freeze on.

A huge amount of work, time, money and will is needed to measure the temperatures within the ice and this data should definitely be published. The authors make a decent effort at clearly presenting data, models and methods. The figures are well thought out and informative. An honest discussion about the limitations in the models and the model set-ups is presented, and I appreciate that.

Nevertheless, I struggled following the line of thoughts. I think the authors should state in a more explicit way, there research questions and how the models can be utilised to find answers to those questions. Especially, which model is able to answer a particular question. I got the impression, that one could come to very similar conclusions running only the '1-D temperature column simulations', but with different forcing fields for the basal mass balance.

Provided Gladstone et al. is published, I could support publication of this manuscript with what I'd call 'minor' revisions.

**Specific comments**

The boreholes exist since several years and a lot of related work has been published as cited in the introduction and section 2.1. Unfortunately, I have difficulty in identifying which temperature-depth profiles are published here for the first time.

All the material regarding the 3-D steady-state temperature simulations with Elmer/Ice heavily depends on Gladstone et al. (2021, in preparation). Although the authors try to summarise the set-up, decisions relevant to this study, and conclusions, this is far from conclusive. E.g.:

- The 'quality' of the simulated 3-D temperature field depends on the inversion quality, and thus information about the missfit between observed and simulated velocity field should be provided (e.g. map of relative velocity difference).

- To what extent do the "further inversions for $\beta$ and $E_\eta^2$" (P. 9, L. 201) change the initial fields shown in Fig. 2?

- I understand, that $\beta$ and $E_\eta^2$ in Fig. 2 are the result of the optimisation in Elmer/Ice to match the observed velocities. Nevertheless, those results appear disconnected from or irrelevant for this manuscript. Further discussion or guidance is needed.

At several places within the manuscript the authors discuss the ice stiffness factor, $B(T_{\rm h})$. Often this seems disconnected from the surrounding text and from my perspective, adds nothing important to the manuscript. I would highly recommend to drop this entirely. Although an Arrhenius law is usually used, "standard physical parameters" (Table A1) do not exist and vary within the literature (e.g. Hooke (1981) versus Paterson and Budd (1982); Paterson (1994)). Even within the same numerical ice-flow model (Elmer/Ice), the choice of rheology parameters $A_0$ and $Q$ (Eq. A1) depends on the model set-up/application (c.f. Gillet-Chaulet et al., 2012; Brondex et al., 2019, Tab. 1). Given the simple algebraic relationship between the ice stiffness factor and temperature, everyone could calculate this according to the specific application.

**Detailed, line-by-line comments, suggestions and technical corrections**

**P. 1, L. 23–26:** Remove "Given the temperature dependence ... input to future modelling studies" — This is only presented in the Appendix Fig. A1.

**P. 4, L. 85:** Reference to Warner at al., 2012 is missing in the "References" section.

**P. 4, L. 96:** "... The temperature profiles ... showed similar profile patterns ( Craven et al. 2009)." — I can't find temperature-depth profiles within the ice in Craven et al. (2004).

**P. 4, L. 107–:** "... with appropriately varying boundary conditions" — That sounds too evaluative to me (non-native speaker). Somewhere in discussion section (4.2.1) the steady-state assumption based on using present-day boundary conditions, even though one particular ice column could travel several hundred years, is mentioned as a model limitation. Probably something like ... *with boundary conditions varying with position.*

**P. 5, L. 133–:** "To eliminate the seasonal signals and derive "steady-state" vertical temperature profiles ..." — From my understanding some sort of temporal averaging was applied to all the datasets listed in Tab. 1. If so, please state this.

**P. 5, L. 137:** "...a multi-year average surface temperature field" — Please state the period. 1979–1998 as on P. 11, L. 254 and P. 13, L. 285+ 305? If the same dataset is used for all the experiments please state this somewhere.

**P. 8, L. 190:** Consider to join the Seroussi et al., 2020 with Greve et al., 2020 to cite the model (Greve) and the specific model application (ISMIP6: Seroussi).

**P. 9, L. 201−:** "We carry out further inversions ..., with different upper and lower surface boundary conditions ..." — I can't identify the use of different upper boundary conditions later in the text or figures. With and without constrained emergence velocity? In Fig. 6 only basal boundary conditions differ.

**P. 13, L. 285:** Any reason not to use skin temperature from the RACMO2.3p2 simulations?

**P. 13, L. 287–288:** "We assume that the grounded ice has zero basal mass balance, while the upper surface is in positive mass balance ..." – Is a positive mass balance also assumed (enforced) at the surface or is the RACMO smb by its own always positive? I do remember that close to the grounding line at AIS the RACMO smb could be slightly negative.

**P. 13, L. 288:** "...the value is extracted from RACMO2.3p2 ..." I assume that a multi-year mean from the RACMO smb has been used here. If so, please state the time period. 1979–2016?

**P. 13, L. 289 and L. 300:** "Melchior Van Wessem et al. 2018" → "Van Wessem et al. 2018"

**P. 13, L. 292–294:** "We thus impose a vertical downward velocity ..." — Is "downward" a direct consequence of the positive smb from RACMO or is this enforced. See also P. 13, L. 287–288.

**P. 13, L. 299:** "...the surface mass balance is extracted from RACMO2.3p2 ..." From the multi-year mean smb or from the RACMO time-series data?

**P. 14, Fig. 4:** This figure mainly duplicates the data presented in Table 3 and I don't think this is worth the space. This figure mainly duplicates the data presented in Table 3, and I don't think this is worth the space. I prefer the authors add the surface and basal temperatures in the figure and drop the table. Keeping just the table would also be acceptable but less illustrative.

**P. 15, L. 21:** "...heat capacity and conductivity, are constant everywhere and will not change with vertical strain process." — According to Tab 4, heat capacity and conductivity depend on the temperature and thus, will evolve over time. Please clarify.

**P. 15, L. 338:** "...the simulations can be evaluated and optimized by comparing the simulated column temperature profile at the borehole sites with borehole measurements." — I don't understand which quantity/process/... is optimized. I thought everything is set now by the choice of model, model parameter and initial and boundary conditions.

**P. 16, L. 366:** Missing space in "Fig.5a" and also in "530 m–624".

**P. 16, L. 368:** "...maintains a hydraulic connection with the ocean below."

**P. 16, L. 369:** Not sure, but "above which" → "where"?

**P. 17, Tab. 5:** Observed temperature gradients and ...

**P. 18, L. 409–411:** "Differences between the simulations and the borehole measurements in the upper surface . . . because the upper surface temperature in the simulations is fixed by the Antarctic surface temperature dataset (Comiso 2000)." — I think this only explains the temperature offset in the upper part of the ice column. The observations at AM05, AM04 and AM01 (Fig. 6) clearly show a temperature profile that is dominated by downward advection near the surface. This is not the case for BMB_CAL2 and only limited for BMB_CAL. I suggest to look into the differences in the vertical velocity (or horizontal flux divergence) at those locations for the different bmb fields. Which model run agrees better with the observed velocity field?

**P. 18, L. 410:** "in the upper surface" → "at the upper surface"?

**P. 20, Fig. 7:** Please extent the x-axis of the insets to align with x-axis of the temperature cross-section. I found it very difficult to relate the basal mass balance to the temperature.

**P. 22, L. 457:** ". . . substantially consistent temperature regime." — I don't understand what is meant here.

**P. 22, L. 476–:** I think this paragraph should go. See 'Specific comments' section.

**P. 22, L. 478–479:** ". . . the depth-averaged flow rate factor is very similar to that of the depth-averaged temperature" — I don't see any reason why this should not exactly correspond to the temperature field (Eq. A1).

**P. 23, L. 490:** "The 1-D . . . higher  resolution in the vertical direction than that from the 3-D simulations." — This sentence is redundant. The 1-D model has only one dimension (vertical) and the model resolutions are stated on P. 13, L. 282 and P. 18, Fig. 6 and again in Fig. 9.

**P. 24, L. 512–513:** ". . . , where basal melting is $7\,\mathrm{m\,a^{-1}}$ and downstream of a region where melt rate exceeds 15 m/a." — Melt rates above 6 m/a are invisible in Fig. 3 due to colour saturation. Please modify Fig. 3 accordingly. Use consistent units ($\mathrm{m\,a^{-1}}$ versus m/a).

**P. 26, L. 587:** ". . . their impact on our steady-state temperature simulations is to impose a non-physical advection of ice through the upper surface." — Please explain, why this would be non-physical? It is already mentioned that those could arise from advecting the geometric features. Wow to distinguish?

**P. 27, L. 602:** "LAGS" or "LAGs"?

**P. 27, L. 606–609:** This is a very long sentence. Consider to split in parts.

**P. 28, L. 652–655:** "Given the differing . . . show marked differences between the four simulations." — This is a very long sentence and I think something got lost.

**P. 29, L. 664–677:** Drop this paragraph. I think it is sufficient to mention the importance of the thermal structure on the rheology and thus on ice dynamics in the Introduction and/or Conclusion.

**P. 30, L. 705–707:** "The depth-averaged temperature-dependent ice stiffness factor, . . . modelling studies of the AIS." — Although the inferred basal resistance parameter and the viscosity enhancement factor could be informative for other model studies, I still don't think that this holds for the stiffness factor. Models could use the temperature field directly.

**References**

Brondex, J., Gillet-Chaulet, F., and Gagliardini, O.: Sensitivity of centennial mass loss projections of the Amundsen basin to the friction law, The Cryosphere, 13, 177–195, https://doi.org/10.5194/tc-13-177-2019, 2019.

Craven, M., Allison, I., Brand, R., Elcheikh, A., Hunter, J., Hemer, M., and Donoghue, S.: Initial borehole results from the Amery Ice Shelf hot-water drilling project, Annals of Glaciology, 39, 531–539, https://doi.org/10.3189/172756404781814311, 2004.

Gillet-Chaulet, F., Gagliardini, O., Seddik, H., Nodet, M., Durand, G., Ritz, C., Zwinger, T., Greve, R., and Vaughan, D. G.: Greenland ice sheet contribution to sea-level rise from a new-generation ice-sheet model, The Cryosphere, 6, 1561–1576, https://doi.org/10.5194/tc-6-1561-2012, 2012.

Goelzer, H., Nowicki, S., Payne, A., Larour, E., Seroussi, H., Lipscomb, W. H., Gregory, J., Abe-Ouchi, A., Shepherd, A., Simon, E., Agosta, C., Alexander, P., Aschwanden, A., Barthel, A., Calov, R., Chambers, C., Choi, Y., Cuzzone, J., Dumas, C., Edwards, T., Felikson, D., Fettweis, X., Golledge, N. R., Greve, R., Humbert, A., Huybrechts, P., clec'h, S. L., Lee, V., Leguy, G., Little, C., Lowry, D. P., Morlighem, M., Nias, I., Quiquet, A., Rckamp, M., Schlegel, N.-J., Slater, D. A., Smith, R. S., Straneo, F., Tarasov, L., van de Wal, R., and van den Broeke, M.: The future sea-level contribution of the Greenland ice sheet: a multi-model ensemble study of ISMIP6, The Cryosphere, 14, 3071–3096, https://doi.org/10.5194/tc-14-3071-2020, 2020.

Hooke, R. L.: Flow Law for Polycrystalline Ice in Glaciers: Comparison of Therretical Predictions, Laboratory Data, and Field Measurements, Reviews of Geophysics and Space Physics, 19, 664–672, 1981.

Paterson, W. S. B.: The Physics of Glaciers, Pergamon Press, Oxford, England, 3 edn., 1994.

Paterson, W. S. B. and Budd, W. F.: Flow parameters for ice sheet modelling, Cold Regions Science and Technology, 6, 175–177, https://doi.org/10.1016/0165-232X(82)90010-6, 1982.

Seroussi, H., Nowicki, S., Payne, A. J., Goelzer, H., Lipscomb, W. H., Abe-Ouchi, A., Agosta, C., Albrecht, T., Asay-Davis, X., Barthel, A., Calov, R., Cullather, R., Dumas, C., Galton-Fenzi, B. K., Gladstone, R., Golledge, N. R., Gregory, J. M., Greve, R., Hattermann, T., Hoffman, M. J., Humbert, A., Huybrechts, P., Jourdain, N. C., Kleiner, T., Larour, E., Leguy, G. R., Lowry, D. P., Little, C. M., Morlighem, M., Pattyn, F., Pelle, T., Price, S. F., Quiquet, A., Reese, R., Schlegel, N.-J., Shepherd, A., Simon, E., Smith, R. S., Straneo, F., Sun, S., Trusel, L. D., Breedam, J. V., van de Wal, R. S. W., Winkelmann, R., Zhao, C., Zhang, T., and Zwinger, T.: ISMIP6 Antarctica: a multi-model ensemble of the Antarctic ice sheet evolution over the 21st century, The Cryosphere, 14, 3033–3070, https://doi.org/10.5194/tc-14-3033-2020, 2020.

---

## Author Comment (AC1)

**Responses to Reviewer #1**

We thank Reviewer #1 for the time and effort spent in reviewing our manuscript in so much detail. The comments are very much appreciated and are of great help in providing an improved version of the manuscript. As itemised below, we will address all the reviewer's specific points in our revisions.

We recognize that the main queries from the reviewer are related to our lack of clarity in descriptions and structure. We reconstructed Sect. 2.2 and Sect. 2.3, presenting the setup and experimental methods of the 3-D and 1-D model. We will detail the revisions of these two sections in the following, together with responses to the specific comments from the reviewer.

**General comments**

The authors present steady-state temperature-depth profiles through the ice at several locations along two distinct flow-lines at the Amery Ice Shelf, East Antarctica, derived from several years of observation within boreholes using thermistor strings and fibre-optical temperature sensing. They further study the sensitivity of simulated steady-state temperature fields on basal mass balance forcing using the full-Stokes ice sheet model Elmer/Ice in a three- dimensional model domain. To add some sort of time dependency, the authors also conduct simulations with one-dimensional 'temperature only' simulations, where the boundary conditions vary along the path and vertical strain is prescribed from the borehole observations.

We have revised the description of the one-dimensional modelling (Sect. 2.3) to clarify that is not immediately concerned directly at incorporating time dependent forcing (although that could be implemented), but rather tracking evolution of the 1-D ice column as it progresses through the current conditions of the ice shelf.

The authors state that the internal temperatures represent a record of the past climate and the thermal conditions upstream, and I would add, that they also serve as a very valuable observable to validate and/or calibrate numerical ice flow models. Especially as ice sheet model assessment is still mainly based on ice sheet wide integral or two-dimensional quantities (Goelzer et al., 2020; Seroussi et al., 2020) ignoring the information that is also preserved in the ice column, e.g., layer depths or temperatures.

The authors find, that basal mass balance forcing derived from remote sensing data leads to the best agreement between the model and the data (given the limitations of the model). I think, this is not surprising as long as ocean models only 'inform' the ice flow model about the basal melt rate without taking the heat flux within the ice towards the ice-ocean interface into account. The same holds for basal melt parametrisations that additionally do not account for basal freeze on.

While we agree that it is not surprising that modelling using the basal mass balance from remote sensing provides the best agreement with the borehole observations, we point out that the basal mass balance from the ROMS ocean model does include regions of freezing (Fig. 3b), although probably not in the appropriate pattern or quantity.

A huge amount of work, time, money and will is needed to measure the temperatures within the ice and this data should definitely be published. The authors make a decent effort at clearly presenting data, models and methods. The figures are well thought out and informative. An honest discussion about the limitations in the models and the model set-ups is presented, and I appreciate that.

We are pleased that the reviewer appreciates the effort involved in obtaining the observations and performing the modelling. As far as we are aware, this is the first comprehensive presentation of temperature observations and 3-D modelling of temperature distribution for a major ice shelf.

Nevertheless, I struggled following the line of thoughts. I think the authors should state in a more explicit way, there research questions and how the models can be utilised to find answers to those questions. Especially, which model is able to answer a particular question. I got the impression, that one could come to very similar conclusions running only the '1-D temperature column simulations', but with different forcing fields for the basal mass balance.

We recognise that the description of our core research questions was not clear enough. We propose that the core research questions are to test the sensitivity of the thermal structure of the AIS to different basal mass balance forcing, and explore the full 3-D temperature field of the AIS. These core research questions are addressed with 3-D modelling. However, in our 3-D temperature simulations, the vertical velocity field is not rigidly prescribed but comes from simulating the dynamics of the ice shelf. In contrast, we directly impose the vertical velocities in the 1-D temperature model and use more accurate upper surface temperatures directly from borehole observations, as a Dirichlet boundary condition. We regard the 1-D model as a complement to our 3-D model.

We revised the last paragraph of the introduction section to emphasis our major research questions, the way to solve them, and the role of the 1-D and 3D models.

Provided Gladstone et al. is published, I could support publication of this manuscript with what I'd call 'minor' revisions.

We recognize that the original manuscript relied too much on Gladstone et al., and it is indeed inappropriate for readers to go to another article to find necessary information about the model used in this study. In the revised 3-D model section, we have added more model information and avoid the dependency of this manuscript on Gladstone et al. The revised manuscript is more self-contained now.

Unfortunately, the full manuscript of Gladstone et al. has been delayed, but details of the aspects of Gladstone et al. (2021, in preparation) relevant to our paper are available in the form of a description of methodology of 3-D inversions on Zenodo to support this manuscript (https://doi.org/10.5281/zenodo.5862046). In the revised manuscript, we cite this as Gladstone et al. (2022) for more detailed model information not covered in this manuscript.

**Specific comments**

The boreholes exist since several years and a lot of related work has been published as cited in the introduction and section 2.1. Unfortunately, I have difficulty in identifying which temperature-depth profiles are published here for the first time.

Thanks for pointing this out. The AM01 and AM04 temperatures have been presented in two papers about the Amery Ice Shelf (Craven et al., 2009; Treverrow et al., 2010), and we have clarified that in our revised manuscript on Line 141 "We note that the temperature profiles of AM01 and AM04 based on thermistor string data have been published in Craven et al. (2009) and Treverrow et al. (2010).".

The temperature-depth profiles of AM02, AM03, AM05, AM06 are published here for the first time, although preliminary results for AM05 and AM06 have previously been presented at conferences. We also modified the text accordingly on Line 382 "Figure 5 shows the borehole temperature profiles at AM01–AM06 grouped as with or without basal marine ice, where the profiles of AM02, AM03, AM05 and AM06 boreholes are published here for the first time. Our time-averaged temperature profiles of AM01 and AM04 boreholes are consistent with the previously published temperature profiles (Craven et al., 2009; Treverrow et al., 2010)."

All the material regarding the 3-D steady-state temperature simulations with Elmer/Ice heavily depends on Gladstone et al. (2021, in preparation). Although the authors try to summarise the set-up, decisions relevant to this study, and conclusions, this is far from conclusive. E.g.:

As we responded above, we agree that the modelling in the original manuscript was highly dependent on Gladstone et al. (2021, in preparation), mainly in terms of the complete 3-D model set-up, inversion processes and the motivations leading to the form of dynamical upper and lower surface boundary conditions we used. Gladstone et al. explores several different aspects of ice sheet and ice shelf modelling, and the present paper is something of an outgrowth of particular approaches to novel boundary conditions for the dynamical equations.

We have made revisions to address the reviewer's specific concerns. In our revised Sect. 2.2, we provide a self-contained summary of the simulations from Gladstone et al., and all the necessary model details has been included.

- The 'quality' of the simulated 3-D temperature field depends on the inversion quality, and thus information about the misfit between observed and simulated velocity field should be provided (e.g., map of relative velocity difference).

We have added a subplot showing the misfit between observed and simulated horizontal surface velocity fields for the starting configuration to Fig. 3 (the original Fig. 2), and added corresponding explanations in the text:
"Figure 3c shows the relative difference in surface horizontal velocities between simulations from the final model state of experiment E3 and observations (Rignot et al., 2017). The relative velocity difference of the ice shelf is basically less than 10%, of which the difference of the fast flow area (where surface velocity >300 m a$^{-1}$) is basically less than 2%. This indicates that the experiment E3 from Gladstone et al. (2022) provides a reliable starting point for the following experiment in this paper."

In the 3-D simulation results section (Sect. 3.2), we also supplement the information for the final velocity misfit:
"The level of agreement of the modelled surface velocities with observations is basically consistent with the result of the E3 (Fig. 3c). The degree of fit for the surface velocity achieved by each of the four BMB experiments is basically the same. For example, in the BMB_CAL experiment, the absolute surface velocity mismatch between simulations and observations (Rignot et al., 2017) of 90% surface nodes on the ice shelf is less than 20 m a$^{-1}$, and the average velocity mismatch is 8.8 m a$^{-1}$. The satisfactory ice dynamic simulation results give us confidence in our temperature simulation results."

- To what extent do the "further inversions for $\beta$ and $E_\eta^2$ (P. 9, L. 201) change the initial fields shown in Fig. 2?

The further inversions of our experiments only slightly changed the parameters shown in Fig. 3 (the original Fig. 2), indicating that the use of the dynamical boundary conditions had little effect on the horizontal flow field. In the 3-D simulation results section (Sect. 3.2), we additionally declare this point:

"The basal resistance and viscosity inversions performed in our ice shelf BMB experiments achieved a very similar distribution pattern to that of the experiment E3 (Fig. 3a, b)."

- I understand, that $\beta$ and $E_\eta^2$ in Fig. 2 are the result of the optimisation in Elmer/Ice to match the observed velocities. Nevertheless, those results appear disconnected from or irrelevant for this manuscript. Further discussion or guidance is needed.

We recognize those results appeared incoherent in the original manuscript, due to the lack of clarity of the previous description of the modelling procedures. β and E$_\eta$ in the original Fig. 2 present the model setup for the starting point of our 3-D simulation. In the revised Sect 2.2, we added clearer guidance, for example:

"The current study uses the final model state of their experiment E3 as our starting point, including the optimised basal resistance parameter β, and viscosity enhancement factor E$_\eta$. The spatial distributions of the two parameters are shown in Fig. 3…"

At several places within the manuscript the authors discuss the ice stiffness factor, $B(T_h)$. Often this seems disconnected from the surrounding text and from my perspective, adds nothing important to the manuscript. I would highly recommend to drop this entirely. Although an Arrhenius law is usually used, "standard physical parameters" (Table A1) do not exist and vary within the literature (e.g. Hooke (1981) versus Paterson and Budd (1982); Paterson (1994)). Even within the same numerical ice-flow model (Elmer/Ice), the choice of rheology parameters $A_0$ and $Q$ (Eq. A1) depends on the model set-up/application (c.f. Gillet-Chaulet et al., 2012; Brondex et al., 2019, Tab. 1). Given the simple algebraic relationship between the ice stiffness factor and temperature, everyone could calculate this according to the specific application.

We regard the discussion of the ice stiffness factor as quite relevant.

However, we do agree with the reviewer that it was incorrect to suggest that there are completely "standard physical parameters" for parameterising the temperature dependence of ice deformation rates, and the corresponding ice stiffness factor $B(T_h)$. We have revised (e.g., in the Appendix) to make it clear that we are simply presenting the results of translating our preferred 3-D temperature distribution into depth-averaged stiffness to comment on the implications for ice dynamics, using the parameterisation (from Paterson 1994) that was used in our Elmer/Ice modelling. This is sufficient to demonstrate how the 3D temperature field influences the ice dynamics. We also changed the term "standard physical parameters" to "parameterized physical parameters" in Table A1.

We also recognise that it was inappropriate to suggest that the distribution of the ice stiffness factor was likely to be an attractive data set for other modellers (compared to the 3-D temperature field), although admittedly some researchers might prefer to commence directly with the 2-D depth-integrated factor. We have proposed what we regard as appropriate changes in emphasis in response to the detailed comments below. However, while to an experienced modeller like this reviewer, there might seem to be little extra information conveyed by the plot of depth-averaged stiffness factor in the Appendix, compared to the depth-averaged temperatures, we feel that (as with Humbert 2010) it is useful to show how the temperature structure affects the pattern of variations in the ice stiffness.

This topic returns at several points in the detailed line-by-line comments below.

e.g., P.1 L. 23-26; P. 14, L476-: P. 29, L. 664–677; P. 30, L. 705–707.

**Detailed, line-by-line comments, suggestions and technical corrections**

**P. 1, L. 23–26:** Remove "Given the temperature dependence . . . input to future modelling studies" — This is only presented in the Appendix Fig. A1.

As discussed above we agree that the last sentence of the Abstract was inappropriate, but there seems to us little to take exception to about the preceding sentence, given that we regard it as helpful to quantify the influence of the temperature distribution on the dynamics. So, we would modify this text to:

"Given the temperature dependence of ice rheology, the depth-averaged ice stiffness factor $B(T_h)$ derived from the most realistic simulated temperature field is presented to quantify the influence of the temperature distribution on ice shelf dynamics. The full 3-D temperature field provides a useful input to future modelling studies."

**P. 4, L. 85:** Reference to Warner at al., 2012 is missing in the "References" section.

Thanks for pointing this out. Added.

**P. 4, L. 96:** ". . . The temperature profiles . . . showed similar profile patterns ( Craven et al. 2009)." — I can't find temperature-depth profiles within the ice in Craven et al. (2004).

Thanks for pointing this out. There is no direct temperature-depth profile in Craven et al. (2004). The original reason Craven et al. (2004) was cited here was because they pointed out for the first time that the lower 150 m of marine ice in the shelf may be an almost isothermal layer, which is related to the statement of profile pattern in the previous sentence. After re-thinking, we agree it is indeed not an appropriate reference here. We deleted it.

**P. 4, L. 107–:** ". . . with appropriately varying boundary conditions" — That sounds too evaluative to me (non-native speaker). Somewhere in discussion section (4.2.1) the steady-state assumption based on using present-day boundary conditions, even though one particular ice column could travel several hundred years, is mentioned as a model limitation. Probably something like . . . *with boundary conditions varying with position.*

Thanks for this good suggestion. Modified to: "… with boundary conditions varying to correspond with position along the flowlines."

**P. 5, L. 133–:** "To eliminate the seasonal signals and derive "steady-state" vertical temperature profiles . . ." — From my understanding some sort of temporal averaging was applied to all the datasets listed in Tab. 1. If so, please state this.

We modified to:
"To eliminate these near surface seasonal signals and derive "steady-state" vertical temperature profiles at the borehole sites for comparison with the simulations, some temperature data near the top surface at AM01–AM04 have been carefully selected and temporal averaged. Temperatures at 10 m depth at AM01 and AM02 are temporally averaged from the collocated AWS records..."

And in the results section of borehole thermal regimes, we have stated that "Borehole thermal regimes are derived from averaging all those measurements (Table 1) selected to represent the thermal equilibrium."

**P. 5, L. 137:** ". . . a multi-year average surface temperature field" — Please state the period. 1979–1998 as on P. 11, L. 254 and P. 13, L. 285+ 305? If the same dataset is used for all the experiments, please state this somewhere.

The period has been added. We added a statement of using the same dataset:

"The surface temperature is determined at each key time stamp (Fig. 4), based on all available AWS observations and multi-year surface mean temperature dataset (the same as used in our 3-D simulations; Comiso, 2000) and is linearly interpolated between time stamps."

**P. 8, L. 190:** Consider to join the Seroussi et al., 2020 with Greve et al., 2020 to cite the model (Greve) and the specific model application (ISMIP6: Seroussi).

Thanks for your suggestion. Changed the text into "a 3-D internal ice temperature distribution generated with the SICOPOLIS model (Greve et al., 2020; Seroussi et al., 2020)".

**P. 9, L. 201–:** "We carry out further inversions . . ., with different upper and lower surface boundary conditions . . ." — I can't identify the use of different upper boundary conditions later in the text or figures. With and without constrained emergence velocity? In Fig. 6 only basal boundary conditions differ.

We recognize that the original statement "We carry out further inversions . . ., with different upper and lower surface boundary conditions" is inappropriate, which is the reason for the reviewer's confusion. In the revised Sect 2.2.2 (Ice shelf basal mass balance experiments), we specify the dynamic boundary conditions for the upper and lower surfaces respectively. We clarify that we used the basal boundary condition with

four different BMB forcings in our study, and a single upper surface boundary condition (a resistive stress scheme from Gladstone et al.). In Fig.6 (the revised Fig. 7), we still present the simulation results of the four different BMB experiments, with borehole observations.

**P. 13, L. 285:** Any reason not to use skin temperature from the RACMO2.3p2 simulations?

Thanks for raising this question. When we set up the 3-D temperature simulation experiments, our focus did not include exploring the response of simulated 3-D temperatures to different surface thermal forcing. Therefore, we only used the surface temperature dataset (Comiso, 2000), which is based on satellite infrared observations.

To explore potential differences between these two temperature data sets, we requested the RACMO2.3p2 skin temperature data and compared it with the satellite derived temperature dataset (Comiso, 2000). As far as the Amery Ice Shelf (AIS) is concerned, we found that the annual mean skin temperatures from RACMO2.3p2 are about 4 degrees colder than those of Comiso (2000). We compared these two surface temperature fields with the temperature observations from the three AWS sites on the AIS and DTS observations from AM05 and AM06, and found that the temperature field of Comiso is significantly closer to those field observations. So based on our further comparison and analysis, RACMO2.3p2 surface temperature forcing is unlikely to bring a better 3-D temperature simulation. As far as the AIS is concerned, the temperature dataset from Comiso (2000) is the better choice.

**P. 13, L. 287–288:** "We assume that the grounded ice has zero basal mass balance, while the upper surface is in positive mass balance . . . " – Is a positive mass balance also assumed (enforced) at the surface or is the RACMO smb by its own always positive? I do remember that close to the grounding line at AIS the RACMO smb could be slightly negative.

The RACMO SMB is indeed slightly negative close to the grounding line of the AIS. We realized that the wording here is unclear. The remark here was intended simply to clarify the vertical velocity profile at the inland start of the flowlines, and that clearly downward surface velocity corresponded to positive SMB. We revised the structure of the description of the 1-D model in Sect. 2.3 to improve the wording and presentation.

We have clarified that we take vertical velocities at each point along the flowline as varying linearly with depth, and determined from the SMB, BMB (taken as zero for grounded ice) and the vertical strain-rate.

**P. 13, L. 288:** ". . . the value is extracted from RACMO2.3p2 . . . " I assume that a multi-year mean from the RACMO smb has been used here. If so, please state the time period. 1979–2016?

Yes, a multi-year mean has been used here. We modified accordingly in the text.

**P. 13, L. 289 and L. 300:** "Melchior Van Wessem et al. 2018" → "Van Wessem et al. 2018"
Corrected.

**P. 13, L. 292–294:** "We thus impose a vertical downward velocity . . ." — Is "downward" a direct consequence of the positive smb from RACMO or is this enforced. See also P. 13, L. 287–288.

Modified in rewriting. This remark was intended to refer to the conditions at the start of the flowline in the grounded ice sheet. We decided that the expression (formerly Equation 4) was not really necessary – and in revising have made it clear that "The vertical velocity in the ice column, taken as varying linearly with depth, is thus determined by the prescribed SMB, BMB and vertical strain rate."

**P. 13, L. 299:** ". . . the surface mass balance is extracted from RACMO2.3p2 . . ." From the multi-year mean smb or from the RACMO time-series data?

We meant "from the multi-year mean smb" here. We have made it clearer by explaining "the surface mass balance (SMB) is extracted from the 1979–2016 mean data of RACMO2.3p2" on Line 335.

While the 1D model following a moving column of ice could be used in conjunction with a coincident temporal time series of forcing, we remind the reviewer that we are considering much longer time intervals than the RACMO period for ice to move though the system, and that we are taking one year time steps.

**P. 14, Fig. 4:** This figure mainly duplicates the data presented in Table 3 and I don't think this is worth the space. I prefer the authors add the surface and basal temperatures in the figure and drop the table. Keeping just the table would also be acceptable but less illustrative.

We dropped the table and as suggested we moved the upper and lower surface boundary temperature conditions at the key points and the vertical strain rates to Fig. 4 (Fig. 5 in the revised manuscript).

**P. 15, L. 21:** ". . . heat capacity and conductivity, are constant everywhere and will not change with vertical strain process." — According to Tab 4, heat capacity and conductivity depend on the temperature and thus, will evolve over time. Please clarify.

Thanks for this comment. We corrected the description here. We clarify that "the ice density of the column is taken as constant everywhere and will not change with vertical strain process. The heat capacity and conductivity are functions of in situ ice temperature."

**P. 15, L. 338:** ". . . the simulations can be evaluated and optimized by comparing the simulated column temperature profile at the borehole sites with borehole measurements." — I don't understand which quantity/process/. . . is optimized. I thought everything is set now by the choice of model, model parameter and initial and boundary conditions.

The upper surface temperature, as a boundary condition, has uncertainties. In the testing phase of the 1-D column simulations, we made some slight adjustments to the upper surface temperatures in the model forcing at those Key Points not directly constrained by AWS observations. We realized it was inappropriate to refer to this as "optimizing" the simulations. We dropped that remark and modified the sentence to "Therefore, the simulations can be evaluated by comparing the simulated column temperature profile at the borehole sites with borehole measurements."

**P. 16, L. 366:** Missing space in "Fig.5a" and also in "530 m–624".
Added.

**P. 16, L. 368:** ". . . maintains a hydraulic connection with the ocean below."

Modified.

**P. 16, L. 369:** Not sure, but "above which" → "where"?
We divide this sentence here into two parts to make it clearer:
"There is a slight step at 530 m depth. Fresher water from the drilling process was observed above this level prior to borehole freeze-up."

**P. 17, Tab. 5:** Observed temperature gradients and . . .
Agreed and modified.

**P. 18, L. 409–411:** "Differences between the simulations and the borehole measurements in the upper surface . . . because the upper surface temperature in the simulations is fixed by the Antarctic surface temperature dataset (Comiso 2000)." — I think this only explains the temperature offset in the upper part of the ice column. The observations at AM05, AM04 and AM01 (Fig. 6) clearly show a temperature profile that is dominated by downward advection near the surface. This is not the case for BMB_CAL2 and only limited for BMB_CAL. I suggest looking into the differences in the vertical velocity (or horizontal flux divergence) at those locations for the different bmb fields. Which model run agrees better with the observed velocity field?

Perhaps the reviewer misunderstood our remark – which we have rephrased to make it clearer that we are referring to the surface temperatures, where the modelled temperature is fixed by the upper surface temperature boundary conditions.

"Differences between the various simulations and the borehole measurements at the upper surface are found at each site, especially for AM05, because the upper surface temperature in the simulations is fixed by the Antarctic surface temperature dataset (Comiso, 2000)."

We note that the reviewer's interpretation that: "The observations at AM05, AM04 and AM01 (Fig. 6) clearly show a temperature profile that is dominated by downward advection near the surface." is perhaps too focused on vertical advection. For example, the near isothermal phenomenon of the AM05 subsurface is not simply caused by downward advection, but more importantly involves horizontal advection of upstream cold ice, as can be seen, for example, in Fig. 7 in the revised manuscript.

As the reviewer suggested, we have looked differences in the vertical velocity at the borehole locations for the different BMB fields. We suggest that, unfortunately, errors in the ice geometry of the model can cause errors in the calculation of horizontal flux divergence, i.e., contaminate the modelled englacial vertical velocity distribution. Therefore, the vertical velocity distribution at the borehole locations cannot be used to evaluate the simulation results. This is one of the limitations of the 3-D model, and we have discussed in detail in our revised manuscript.

Regarding the reviewer's last point "which model run agrees better with the observed velocity field", we suggest that comparing the velocity field mismatch of the four BMB experiments cannot be used to evaluate the BMB fields. The effect of different BMB forcings on the mismatch between the simulated velocity field and the observations is minimal.

**P. 18, L. 410:** "in the upper surface" → "at the upper surface"?
Modified.

**P. 20, Fig. 8:** Please extent the x-axis of the insets to align with x-axis of the temperature cross-section. I found it very difficult to relate the basal mass balance to the temperature.

Thanks for this good suggestion. We re-plotted this figure as suggested to make it clearer.

**P. 22, L. 457:** ". . . substantially consistent temperature regime." — I don't understand what is meant here.

What we mean is that downstream of AM05, the vertical temperature structure of the ice shelf is stable (Fig. 8a); i.e., the simulated vertical temperature distributions of AM05, AM04, and AM01 are similar.

Our remarks were a little over-simplified, since the BMB_CAL forcing suggests substantial variations in the basal freezing rate along the flowline downstream of AM05. We meant that downstream of AM05, the vertical temperature structure of the ice shelf on this flowline maintains a consistent profile (Fig. 8a). The simulated (and observed) vertical temperature profiles at AM05, AM04, and AM01 (Fig. 6 a-c) are similar, essentially just scaling as the ice shelf thins despite the continuing accretion of marine ice.

**P. 22, L. 476–:** I think this paragraph should go. See 'Specific comments' section.

As we responded above under Specific Comments, we disagree with the reviewer about the value of pointing out the actual implications for the ice shelf flow of the temperature distribution, through the depth-averaged ice stiffness factor, B. Our motivation is essentially similar to that of Humbert (2010) who presented a similar figure.

We reconstruct this paragraph and supplement the interpretation of the pattern of $B(T_h)$:
"As the ice keeps progressive warming along the flowlines downstream (Fig.7), the depth-averaged $B(T_h)$ decreases, corresponding to the softening of ice, while there is also significant lateral variation in stiffness across the ice flow."

**P. 22, L. 478–479:** ". . . the depth-averaged flow rate factor is very similar to that of the depth-averaged temperature" — I don't see any reason why this should not exactly correspond to the temperature field (Eq. A1).

There is obviously a strong resemblance, as stated, but some moment's reflection should bring the realisation that the ice stiffness factor evaluated using the depth-averaged temperature $B(\overline{T_h})$ is not directly comparable to the depth-average of the function factor $B(T_h(z))$. We retain the interpretation here.

**P. 23, L. 490:** "The 1-D . . . higher vertical resolution in the vertical direction than that from the 3-D simulations." — This sentence is redundant. The 1-D model has only one dimension (vertical) and the model resolutions are stated on P. 13, L. 282 and P. 18, Fig. 6 and again in Fig. 9.

We agree and the sentence has been dropped.

**P. 24, L. 512–513:** ". . ., where basal melting is 7 m a$^{-1}$ and downstream of a region where melt rate exceeds 15 m/a." — Melt rates above 6 m/a are invisible in Fig. 3 due to colour saturation. Please modify Fig. 3 accordingly. Use consistent units (m a$^{-1}$ versus m/a).

Thanks for pointing this out. We have checked for the consistency of all units abbreviations.
We note the reviewer's concerns about Figure 3 and the saturation of the very high melt-rates in the southernmost part of the ice shelf. However, peak melt rates exceed 45 m a$^{-1}$ and we do not want to lose the resolution of the low melt regions – already hard enough to discern for the ISMIP6 case in Figure 3a. We have devised a nonlinear colour palette to significantly reduce the saturation. In the new palette, the colour is saturated at 15 m a$^{-1}$.

**P. 26, L. 587:** ". . . their impact on our steady-state temperature simulations is to impose a non-physical advection of ice through the upper surface." — Please explain, why this would be non-physical? It is already mentioned that those could arise from advecting the geometric features. How to distinguish?

We recognise the discussion of the emergence velocity here in the original manuscript was not clear enough. We have conducted proper interpretation and discussion of the upper surface emergence velocity in the revised manuscript.

In short, the high emergence velocities from the inversions (here "high" means higher than could be explained through thinning rates and/or SMB) might represent physically plausible advection of geometric features, or they might be pure artifacts (for example due to bedrock uncertainty). We don't have a good way of distinguishing between the two.

**P. 27, L. 602:** "LAGS" or "LAGs"?

We modified all LAGs to LAGS.

**P. 27, L. 606–609:** This is a very long sentence. Consider to split in parts.
We agree that this was a rather involved sentence, attempting to cover several points at once. We have revised as follows:

"The Antarctic bedrock is difficult to observe with spatially consistent accuracy. Even ice shelf thicknesses are not well constrained, because they are often derived assuming local hydrostatic equilibrium (particularly where marine ice accretion prevents direct radar measurements). This means that very high accuracy upper surface elevations are required in order to infer the ice draft. Lack of detailed ice density profile data also contributes to uncertainties in the buoyancy calculations."

**P. 28, L. 652–655:** "Given the differing . . . show marked differences between the four simulations." — This is a very long sentence and I think something got lost.

This was intended simply as an introduction to the remarks contrasting the various modelled temperatures, as judged by the comparison with observations at the marine ice sites. Perhaps it was also inappropriate to be "not surprised" with the benefit of hindsight, that the different choices of basal mass balance forcing were sufficient to produce significantly different temperature distributions.

We revised as follows:
"Our simulations explored different distributions of basal mass balance, shown in Fig. 3. The comparisons between simulated and measured temperature profiles in marine ice locations (Fig. 6a, b, c) showed marked differences between the four simulations."

**P. 29, L. 664–677:** Drop this paragraph. I think it is sufficient to mention the importance of the thermal structure on the rheology and thus on ice dynamics in the Introduction and/or Conclusion.

First, as we discussed under Specific Comments, we think this emphasis on the ice shelf flow is relevant. Second, we hold to the general principle that one does not introduce new material in the Conclusions.

Accordingly, we think this paragraph belongs in the Discussion section. It puts our results in the context of previous work. Furthermore, it contains relevant remarks about additional uncertainties concerning the mechanical properties of the marine ice layers. Consistent with our earlier remarks, we have deleted the sentence about "The 3-D field of the ice stiffness factor … and future ice dynamic modelling studies of the AIS".

**P. 30, L. 705–707:** "The depth-averaged temperature-dependent ice stiffness factor, . . . modelling studies of the AIS." — Although the inferred basal resistance parameter and the viscosity enhancement factor could be informative for other model studies, I still don't think that this holds for the stiffness factor. Models could use the temperature field directly.

Consistent with our response about the discussion of the ice stiffness factor, we see the first sentence here: "The depth-averaged temperature-dependent ice stiffness factor $B(T_h)$ of the AIS is also calculated from the BMB_CAL temperature field to quantify the dependence of ice viscosity on temperature and demonstrate its influence on dynamics."

As conventional in Conclusions, reiterating what was presented in the paper. We consider that standard practice. Again – deferring to the reviewer's concerns about the range of options available for modelers to implement ice flow rates (or stiffness factors), we have altered the following sentence to read: "The 3-D temperature field can be used for further dynamic modelling studies of the AIS."

---

## Author Comment (AC2)

**Responses to Reviewer #2**

In this manuscript, the authors present observational and modelling results for the temperature distribution of the Amery Ice Shelf. Observations are from borehole measurements along three different flow lines and encompass sites with and without basal layers of marine ice. Two types of simulations are discussed, namely (i) 3-D simulations with the Elmer/Ice model, and (ii) 1-D simulations of ice columns advected along flow lines. The authors compare the findings from the observations and simulations, revealing a generally reasonable agreement. Systematic deviations occur in the marine ice layers, for which the observations show essentially isothermal conditions due to the presence of liquid saltwater. Reproducing this by the simulations would require a particular treatment of the thermodynamics of marine ice, which is identified as a crucial component for future work.

The study is somewhat preliminary in nature, and I do not think it has the potential to be a game-changer in the field. However, science is largely an evolutionary process, and not every paper can be. It is still a decent piece of work and presents results that are of interest to the community. Therefore, I think it should be considered for publication. However, I would like to raise some issues that should be dealt with before that.

Clearly this is not the last word in simulating the 3-D temperature distribution in the Amery Ice Shelf. We were pleased that the reviewer recognised that our study highlights the need for a more sophisticated treatment of the thermodynamics of marine ice zones. We naturally agree that the results should be of interest to the community. To the best of our knowledge, this is the most comprehensive comparison of modelled and observed ice shelf temperatures presented to date.

Since the paper draws strongly on Gladstone et al. (2021), I think it should not be published before the latter one is available at least as a preprint with a persistent identifier (e.g., DOI). The separation of the two studies does not always become clear. This refers in particular to Section 2.2: I don't fully understand what was already done by Gladstone et al. (2021), and what is new. This should be made crystal clear in all details. Further, as Section 2.2 is a subsection of "Data and methods", any results (such as Fig. 2 and accompanying text - even if already discussed by Gladstone et al. (2021)) should be moved to Section 3.

The paper we cited in our original manuscript as Gladstone et al. (2021, in preparation) has been delayed, and we have taken up the reviewer's suggestion as much as possible by making the detailed methodology of that work available and citable as Gladstone et al, (2022) on Zenodo (https://doi.org/10.5281/zenodo.5862046).

We recognize that the original Sect. 2.2 was not clear enough, and the boundary between this manuscript and Gladstone et al. (2021, in preparation) was a bit vague. We have reconstructed the structure of Sect. 2.2 to make clearer the relationship between Gladstone et al. (2022) and this manuscript.

"The simulations presented here build on a larger study currently in progress (methodology described by Gladstone et al., (2022)), and we take simulations from that work as our starting point…"

We also provide more details of our model set-up in Sect. 2.2 to make it self-contained.

To summarise the methodology document:

Gladstone et al (2022) describes exploration of several approaches to initialising regional Antarctic ice sheet models, with respect to optimisation of parameters controlling basal sliding and ice stiffness, via the basal resistance parameter $\beta$, and the viscosity enhancement factor $E_\eta$, respectively. It also examines alternative boundary conditions for the solution of the dynamical equations (the Stokes or momentum balance equations).

Our paper uses several of those concepts. In particular, by prescribing the basal mass balance

of the ice shelf as part of the dynamical ice shelf basal boundary conditions, as suggested by Gladstone et al. (2022), we are able to explore how different estimates of the distribution of basal melting and freezing for Amery Ice Shelf affect the 3-D ice velocity field and thereby influence the 3-D temperature distribution throughout ice shelf. The other novel feature used in our paper is that, again based on the experience of studies in Gladstone et al. (2022), we adopted a different dynamical boundary condition at the upper ice surface. We have explained both these non-standard boundary conditions in more detail in our revised Sect. 2.2.2.

Regarding the presentation of the optimized parameter fields for our starting point, in Figure 2 (Figure 3 in the revised version), we disagree with the Reviewer's contention that these belong in Results. They are not results of our work, but inputs. In our revised Sect. 2.2.1, we further clarify this point:

"The current study uses the final model state of their experiment E3 as our starting point, including the optimized basal resistance parameter $\beta$, and viscosity enhancement factor $E_\eta$. The spatial distributions of the two parameters are shown in Fig. 3, together with the achieved match to observed velocities (Rignot et al., 2017)."

The story with the temperature field of the 3-D simulations confuses me. In line 165, the authors say that the temperature field is computed by Elmer/Ice, whereas in lines 188-190, they explain that they use englacial temperatures computed by SICOPOLIS for ISMIP6-Antarctica. The latter statement is essentially repeated in lines 198/199 ("The original ice temperature field from the SICOPOLIS modelling is retained throughout"). How does this go together?

We apologise for the confusion, and have revised in the new Sect. 2.2 to clarify this:

"In our simulations, we first optimize the ice flow dynamics across the LAGS for each of our choices for ice shelf BMB forcing. We do this by optimizing spatial distributions of basal resistance and ice viscosity using adjoint inverse methods..." And "we use a 3-D ice temperature field from the SICOPOLIS modelling (Greve et al., 2020; Seroussi et al., 2020) throughout the two inversion steps (Fig. 2)."

After these two inversion steps, "we then generate the corresponding 3-D steady-state temperature distributions, using Elmer/Ice to solve the steady-state advection-diffusion equation..."

I noticed quite a few issues with the English writing. Some of them are pointed out below, but the manuscript can certainly do with a very thorough round of proofreading.

Thanks for your suggestions below. We have read the full text repeatedly and conducted a very thorough proofreading. Several authors have also reviewed the entire manuscript as suggested.

**Detailed comments:**

Line 33/34: Inconsistent capitalization. I'd suggest capitalizing the whole term as "Lambert-Amery Glacial _S_ystem (LAG_S_)". This entails several further changes to "LAG_S_" throughout the manuscript.

Thanks for pointing this out. We agree to use the term LAGS throughout the manuscript.

Line 50: "of _the_ marine ice layer"

Modified.

Line 56: "in_-_situ"

We checked the English guidelines of The Cryosphere. It said that *Latin phrases should not be hyphenated (e.g., "in situ", not "in-situ").*

Line 64: "water_-_filled"

Modified.

Line 84: "in_-_situ"

As we responded regarding Line 56, *Latin phrases should not be hyphenated (e.g. "in situ", not "in-situ") as required by the Cryosphere.*

Line 91: of _the_ basal melt rate

Modified.

Line 94: "lower surfaces_,_ and the transition temperature"

Added.

Line 120: "through _the_ AM03 borehole"

Added.

Line 121: "close by _the_ AM02 borehole"

Added.

Line 124: "ice shelf temperature_,_ and the other"

Added.

Line 163:"The 3-D steady-state temperature simulations"->"3-D steady-state temperature simulations"

Modified.

Line 166: "ice flow dynamics_,_ the current study"

Added.

Equation (1): Add a comma after the equation.

Added.

Line 171: "varies the viscosity, η" -> "varies the viscosity η"

Modified.

Equation (2): Add a comma after the equation.

Added.

Equation (3): Add a full stop after the equation.

Modified.

Equation (4): Why is the enhancement factor squared? Usually (e.g., Greve and Blatter, 2009, "Dynamics of Ice Sheets and Glaciers", Springer, Sect. 4.3.4), it appears as a linear term in the ice viscosity.

In Greve and Blatter (2009), their viscosity expression is actually

$$\eta = \frac{1}{2} E^{-1/n} A(T_h)^{-1/n} \dot{\varepsilon}_e^{\frac{(1-n)}{n}}$$

because they term $E$ a *flow enhancement factor* for the Glen flow relation, connecting strain rates to deviatoric stresses (essentially a prefactor to the *flow rate factor* $A$). So, the relationship between their factor and our *viscosity enhancement factor* is:

$$E^{-1/n} = E_\eta^2$$

In the inversion process, the parameter we optimise doesn't have to be the flow enhancement factor ($E$) itself. We use the square of our optimizing parameter to ensure that the factor rescaling the viscosity is never negative which would be non-physical. This is about the separation between the optimization process itself and the actual enhancement factor in the ice flow relation, as used to calculate the effective viscosity. In a similar fashion, the use of the dimensionless parameter β to parameterize the basal drag coefficient ensures that the coefficient is always positive so that basal friction always opposes the sliding velocity.

Equation (4): Add a full stop after the equation.

Added.

Line 188: "In this study_,_ we utilize"

Added.

Line 190:"generated with the SICOPOLIS model (Greve et al., 2020; Seroussi et al., 2020)"

Modified.

Line 210, "Tikhonov regularisation parameters": Give a reference for this.

Thanks for this comment. We added two related references here.

Equation (5): Add a comma after the equation.

Added.

Lines 259/260: "We use _a_ surface resistance coefficient..., and _a_ reference speed..."

Modified.

Line 290: "is assumed _to be_ given by"

Modified.

Equation (6): Add a comma after the equation.

We removed Equation (6) after rewriting the 1-D simulation descriptions.

Lines 293/294: "scale it linearly with depth to zero at the lower surface to approximate the vertical advection": While this should be a reasonable approximation for grounded ice, I don't think it is good for floating ice because of the generally significant basal mass balance (melting/freezing). This requires a comment.

Thanks for drawing attention to a source of confusion. This original remark was meant to apply only to the grounded ice sheet. We have substantially rearranged the description of the 1-D column simulations (Sect. 2.3) and improved the wording.

We clearly present the SMB and BMB used throughout the 1-D simulation process "According to the location of the ice column at each time step, the surface mass balance (SMB) is extracted from the 1979–2016 mean data of RACMO2.3p2 (a regional atmospheric model; Van Wessem et al., 2018). Similarly, the BMB within the floating sector is extracted from Adusumilli et al. (2020) (i.e., BMB_CAL), while zero BMB is imposed for the grounded ice." And we clarified that "The vertical velocity in the ice column, taken as varying linearly with depth, is thus determined by the prescribed SMB, BMB and vertical strain rate."

Line 313: "column simulation_s_"

Modified.

Line 335: "at _the_ pressure melting point "

Modified.

Line 344: "Borehole thermal regime_s_"

Modified.

Fig. 5b: What is the meaning of the dotted parts of the red curve?

We have explained this in the last sentence of the caption of Fig. 5 (Fig. 7 in the revised manuscript).

Line 357: "(a) at AM01, AM04 and AM05, and (b) at AM02, AM03 and AM06" -> "(a) at AM01, AM04 and AM05 (with marine ice), and (b) at AM02, AM03 and AM06 (without marine ice)"

Thanks for this suggestion. Modified.

Line 359: "mark impermeable (solid line) _and_ permeable (dotted line)"

Modified.

Line 360: of _the_ upper and lower surfaces

Added.

Line 370: "500m" -> "500 m"

Modified.

Lines 452/453: "from grounded ice to floating ice shelf" -> "from grounded to floating ice"

We wanted to emphasise the different regimes clearly, so we have replaced with "from grounded ice sheet to floating ice shelf".

Line 510: "at _the_ AM02, AM03 and AM06 sites"

Modified.

Line 536: "at high-elevation" -> "at high elevation"

Thanks for pointing out the stray hyphen. We replaced this with "at high elevations".

Line 545: "with _the_ ocean below"

Added.

Lines 581/582, "Considering that it takes ~1100 years for ice to reach the ice front from the southern grounding zone": This statement requires a reference.

We modified this to read: "Considering that it takes ~1100 years for ice to reach the ice front from the southern grounding zone under present day velocities (Rignot et al., 2017), this could cause a bias in our simulated temperature distribution."

Line 583: "20th _c_entury"

Modified.

Line 584: "steady state_,_ then"

Added.

Line 586: "spatially variations" -> "spatial variations"

Modified.

Line 608: "measurements)_,_ which means"

Thank you. Resolved when we rewrote this long sentence (lines 696-610) in response to Reviewer 1.

Line 632: "in_-_situ"

As we responded for Line 56, _Latin phrases should not be hyphenated (e.g. "in situ", not "in-situ") as required by the Cryosphere._

Lines 694-697: These two sentences are a bit convoluted. Rather something like "The AM01, AM04 and AM05 boreholes have a permeable basal layer of porous marine ice approximately 100 m thick, which appears to conform to the pressure-dependent seawater freezing temperature. The AM02, AM03 and AM05 boreholes experience active melting, and large temperature gradients up to -0.36 degC m-1 are found at the base."

Thanks for this suggestion, modified.

Line 721: "This study presents the first quantitative analysis of the 3-D temperature field of the Amery Ice Shelf":

Is this really true in view of the companion paper by Gladstone et al. (2021, in prep.)?

As mentioned earlier, Gladstone et al (2021, in preparation) has been delayed. That paper demonstrates the sensitivity of ice shelf temperatures to the different choices of different upper surface dynamic boundary conditions, but this is certainly the first quantitative analysis for the Amery Ice Shelf capable of discriminating between different temperature distributions by comparison with experimental observations.

Line 730: "of _the_ depth-averaged"

Modified.

Line 733: Put the divisor "(RT_h)" in brackets.

Modified.

Lines 824/825: This reference is wrong. It should rather be

"Greve, R., Calov, R., Obase, T., Saito, F., Tsutaki, S. and Abe-Ouchi, A.: ISMIP6 future projections for the Antarctic ice sheet with the model SICOPOLIS, ..."

Thanks for pointing this out. Modified. In addition to this, we have thoroughly checked all the references for errors.

---

## Author Response (AR1)

为获得最佳体验，请在 **Acrobat X、Adobe Reader X 或更高版本**中打开此 **PDF** 包。

立即下载 **Adobe Reader**！

---

## Author Response (AR2)

**Responses to editor's comments**

We thank the editor for careful review and the following comments.

L31: I find the acronym AIS for Amery Ice Shelf quite inconvenient given that it is the same one used for Antarctic Ice Sheet. I suggest considering something like "AmIS" or "AMIS" to disambiguate.

To the best of our knowledge, "AIS" has been used for Amery Ice Shelf within almost all Amery papers for a long time (e.g., Craven et al., 2009; Galton-Fenzi et al., 2012; Herraiz-Borreguero et al., 2013). We thought it was a bit confusing to propose a new acronym in this paper, so we would like to continue using the acronym "AIS" from the previous Amery papers.

L510: dynamical simulations => dynamic simulations

We thought dynamic is used to describe something is changing (i.e., non-static), while "dynamical" refers to something involving dynamics. We thought "dynamical simulations" (as distinct from thermal simulations) would be a slightly better expression for "simulations of ice dynamics" here.

L696: dynamical regime => dynamic regime

Thanks for this suggestion. We thought "dynamics" would be a more direct expression in this case. We would modify to:

"In general, the thermal structure of ice shelves influences ice rheology and therefore also the  dynamics (Humbert, 2010; Budd & Jacka, 1989)."

Correspondingly, we also modified the wording at L788,

"It is close to zero over the ice shelf, and thus has very little impact on the  dynamics within the ice shelf."

We also noticed in passing that we have some inconsistency in using "dynamic" or "dynamical" when referring to the boundary conditions. For consistency, we used "dynamic boundary condition" throughout the manuscript.